# Large-scale capture of hidden fluorescent labels for training generalizable markerless motion capture models

Daniel J. Butler [1], Alexander P. Keim [1], Shantanu Ray[1] & Eiman Azim [1] ✉

Deep learning-based markerless tracking has revolutionized studies of animal behavior. Yet the generalizability of trained models tends to be limited, as new training data typically needs to be generated manually for each setup or visual environment. With each model trained from scratch, researchers track distinct landmarks and analyze the resulting kinematic data in idiosyncratic ways. Moreover, due to inherent limitations in manual annotation, only a sparse set of landmarks are typically labeled. To address these issues, we developed an approach, which we term GlowTrack, for generating orders of magnitude more training data, enabling models that generalize across experimental contexts. We describe: a) a high-throughput approach for producing hidden labels using fluorescent markers; b) a multi-camera, multi-light setup for simulating diverse visual conditions; and c) a technique for labeling many landmarks in parallel, enabling dense tracking. These advances lay a foundation for standardized behavioral pipelines and more complete scrutiny of movement.

One of the key challenges in studying how the body is controlled as it interacts with the environment is simply measuring movement in the first place. Historically, movement studies have employed a wide variety of measurement techniques. Perhaps the earliest and most widely used strategy is qualitative observation, where investigators watch the animal and describe the movements they see[1,2]. This approach has the advantage of not impinging on the subject's behavior, while allowing the investigator to judge which aspects of a complex set of movements are relevant. By the same token, however, human observation is subject to limitations and biases – the observer may not discern subtle features of a movement or may unwittingly fail to record the features relevant to the question being asked.

An important addition to the scientific toolbox was the introduction of marker-based tracking. With this technique, markers are attached to the animal and marker positions are computed from video cameras or other sensors[3–8]. Marker-based approaches afford high accuracy but have several drawbacks. Some animals do not tolerate markers well, and particularly with smaller animals like rodents, it can be difficult to place markers on fine structures like the digits. Furthermore, marker-based tracking is still subject to a degree of human

bias, as the investigator must choose where to place the markers well before collecting and analyzing the data.

More recently, as deep learning[9,10] has become a mature technology for extracting information from images and video, markerless tracking methods (e.g., DeepLabCut[11], LEAP/SLEAP[12,13], and DeepPoseKit[14]) have become widely adopted[15–17]. These markerless tracking systems are often used with the following workflow: (1) the investigator records a behavior of interest and manually labels landmarks in a subset of captured video frames; (2) a deep learning model is trained on these labeled frames; (3) the model is used to predict landmark positions in the other frames that have not been manually labeled; (4) the investigator verifies the labels produced by the model; and finally, (5) if errors are sufficiently rare, analysis proceeds; otherwise the investigator returns to step (1), labels more frames, and trains and applies a new deep learning model with the expectation that a larger training set will reduce the error rate to an acceptable level.

Markerless tracking based on this workflow can produce highly accurate results, sometimes with as few as 100 to 1000 labeled training images. It also carries the advantage that it can be applied post-hoc to video data that was collected without tracking in mind, making it

---

[1]Molecular Neurobiology Laboratory, Salk Institute for Biological Studies, 10010 N. Torrey Pines Road, La Jolla, CA 92037, USA. ✉e-mail: eazim@salk.edu

possible, for instance, to quantify the movement of animals in their natural habitats from archival footage. Already, the workflow described above has had a major impact on a variety of fields that study animal behavior, and the applications continue to grow.

Our approach builds on this influential work in markerless tracking and aims to overcome several key obstacles. One obstacle is poor robustness. Although deep learning models trained using the process described can in principle be applied to new experimental setups and visual environments not seen during training, in practice, even small changes, for example in lighting, camera position, or the behavior being measured, often cause accuracy to drop dramatically. Therefore, new setups require new images to be manually labeled and new models to be trained. Another obstacle is that the manual labeling of landmarks has inherent limitations. For one, only landmarks that can be easily identified by humans across images can be labeled. Moreover, the need for human labor increases time and cost, and accuracy is limited by skill level and consistency across human annotators.

Several approaches have recently been proposed for training motion capture models for scientific applications without the need for manual labeling of newly collected data. Some notable methods rely on multi-camera-based 3D reconstruction[18], implanted physical markers[19], or a pre-built 3D model of the target object[20]. We sought to develop an alternative strategy that is non-invasive, does not rely on 3D reconstruction or 3D models, and can enable automated, high-density labeling of the subject. To address these challenges, we developed GlowTrack, an approach to automatically generate large, diverse image datasets along with corresponding landmark labels without the need to manually label video frames. To collect these large training sets, we label the subject with a fluorescent dye that is only excited by wavelengths outside the visible range, and alternately strobe excitation and visible light sources to collect pairs of images in which the dye is either visible or absent. Image data is then processed and used to train a model that can detect landmarks on animals that do not have any fluorescent label applied. Primarily due to the dramatic increase in data set size and visual diversity, deep learning models trained with this approach are robust enough to operate in different contexts, giving experimenters more flexibility to change their setups without labeling additional images or training new models.

We present two pipelines for turning fluorescent labels into trained landmark trackers. The serial labeling pipeline enables reliable tracking of specific landmarks of interest. The parallel labeling pipeline enables tracking of an arbitrary number of automatically selected landmarks, eliminating the need for the investigator to pre-select a small number of landmarks a priori, thereby reducing bias and increasing coverage (Fig. 1). Ultimately, more sensitive, reliable, and comprehensive methods to capture and quantify movement could have wide-ranging use in experimental science, ethology, clinical diagnosis, robotics, and augmented reality.

## Results

Our approach to generating labeled images is based on applying hidden fluorescent fiducials to body regions of interest, inspired by the use of fluorescent dye to generate ground truth data for other applications[21]. Here, we establish the hidden fluorescence approach for deep learning-based markerless tracking and develop several innovations to optimize it for generating high-quality labels at large scale. We describe two different variants of our labeling approach, each tuned for different performance characteristics (Fig. 1).

The first variant, which we call serial labeling, can be used to label landmarks on different surface types, even textured surfaces with hair (Fig. 1a). In the serial labeling approach, a single landmark is labeled with the fluorescent dye, and the data synthesis process is repeated for each landmark one wishes to label. As proof-of-concept, we use the serial labeling procedure to train a robust deep learning model for tracking the hand of the mouse and evaluated its accuracy on

challenging image data from novel experimental setups and behaviors. We further improve performance by using real-time feedback from the model to optimize camera settings and achieve better accuracy.

The second variant, which we call parallel labeling, expands this concept to the labeling of an unbounded number of landmarks simultaneously (Fig. 1b). In the parallel labeling approach, dye is applied in a random speckle pattern, yielding hundreds to thousands of visually identifiable local regions, or visual barcodes (in analogy to the synthetic nucleic acid barcodes used in biological applications[22–24]), that can be matched across images. We apply this procedure to the human hand, training a deep learning model to track many landmarks in parallel at a density of coverage decided by the user. Our approach is not restricted to tracking the human hand, an active research area with an extensive literature[25–28]; rather, we use the hand as proof-of-concept that our data generation technique can be used to quickly train a versatile model that tracks many points on a complex, articulated object.

The two variant approaches share many attributes in common, so for clarity of exposition, we first describe serial labeling and associated experiments, and then describe parallel labeling as an extension.

### Serial labeling with hidden fluorescent dye

In the serial labeling pipeline (Fig. 1a), a single target landmark is marked with an ultraviolet (UV) fluorescent dye using a fine brush or felt-tipped marker. We use UV fluorescent dye because of its commercial availability, but, in principle, dyes that fluoresce at other wavelengths could be used. UV and visible illumination sources are strobed alternately, switching the dye between fluorescing and invisible states (Supp. Movie 1). The centroid of the dye region is then computed from the UV-illuminated images in post-processing (Supp. Fig. 1). The dye centroid from each UV frame is used as a proxy label for the subsequent visible frame in the video, thereby transferring each UV-computed centroid to a visible image. This transfer procedure is justified because, at high video frame rates, the discrepancy between the landmark locations in adjacent UV and visible frames is negligible (Fig. 2a and Supp. Fig. 1; the frame rate was 200 Hz for the experiments reported here, well within the range of standard machine vision cameras). In this way, a large, diverse, and accurately labeled image dataset can be generated without the need for manual labeling of any frames. In the final step of the pipeline, this large dataset undergoes a process of data augmentation (see Methods) while it is used to train a deep learning model that can be used to predict landmark location in novel images in which no dye is present (Supp. Fig. 1).

The relative timing of the illumination phases and image capture is important. In the simplest implementation of serial labeling, which we call biphasic illumination, each fluorescence image is captured while the UV source is actively illuminating the dye, after which the illumination is switched to visible light for visible image capture (Fig. 2b, top). Although this scheme is intuitive, the UV source is on during image capture, allowing light from the UV source to leak into the camera, decreasing the signal-to-noise ratio and making the dye more difficult to localize during post-processing (Fig. 2b, bottom). Mounting an optical low-pass filter onto each camera reduced background signal from the UV source but did not completely eliminate it, likely because the UV LEDs emit a broad spectrum that includes some wavelengths in the visible range.

A more serious problem that could not be solved with spectral filtering was the natural fluorescence of objects in the scene (Fig. 2b, bottom). To eliminate natural scene fluorescence, we developed a second image capture configuration that we call triphasic illumination. This scheme is based on our observation that the natural fluorescence in the scenes we captured had a short temporal decay constant (likely on the order of less than 1 microsecond). We reasoned that if we use a dye with a longer temporal decay constant, the UV source can be turned off just prior to image capture, allowing natural fluorescence to

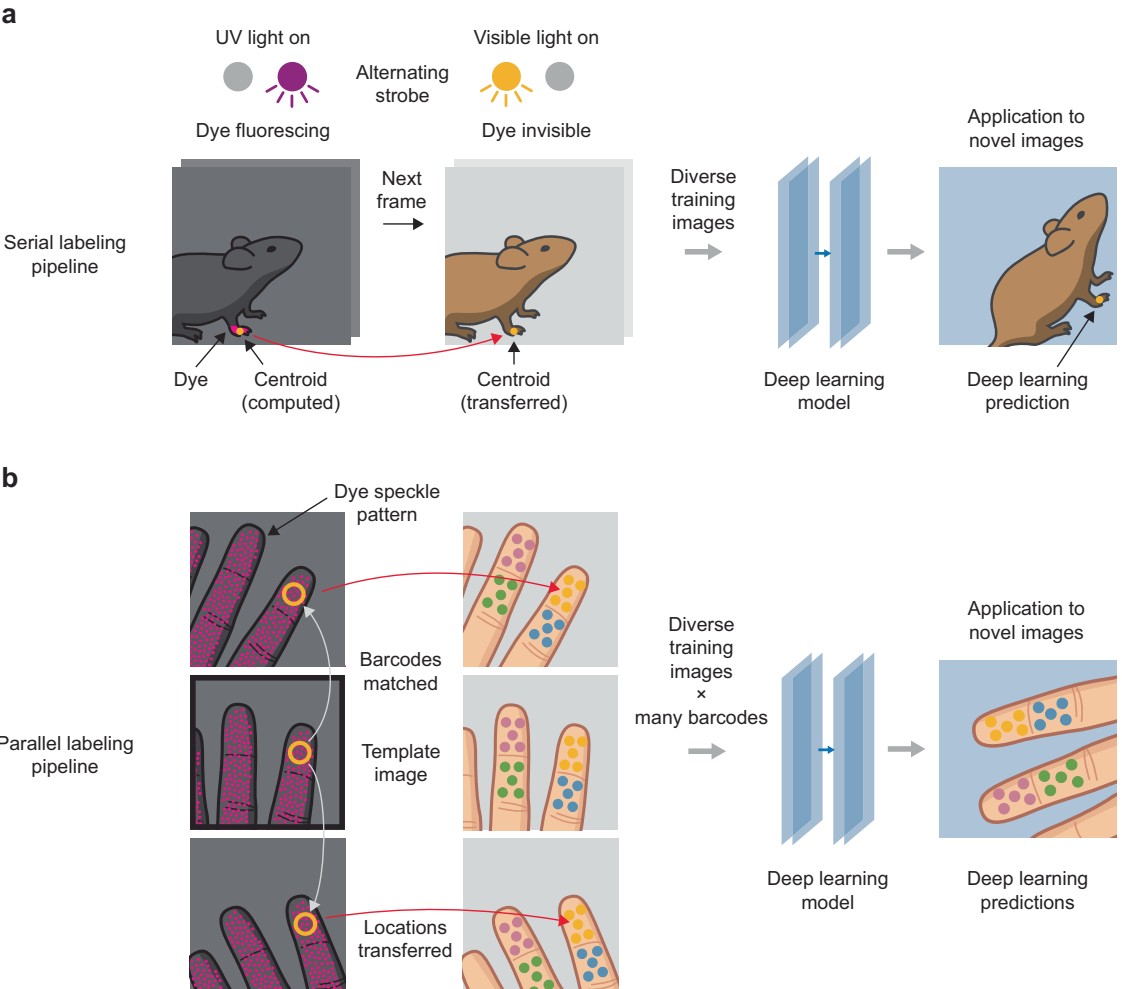

**Fig. 1 | Hidden fluorescent labels for training versatile landmark detectors. a** In the serial labeling pipeline, a target landmark is marked with a fluorescent dye. UV and visible illumination sources are strobed alternately, switching the dye between fluorescing and invisible states. The centroid of the dye region is computed in post-processing and is used as a landmark label for the subsequent visible light frame (red arrow), in which dye is invisible (see Supp. Fig. 1). In this way, a large, automatically labeled image dataset is generated and used to train a deep learning model to predict landmark locations in novel images. **b** In the parallel labeling pipeline, many landmarks are labeled simultaneously with a fluorescent speckle pattern where local neighborhoods form distinctive visual barcodes. A small number of images are selected as templates (black outline), and labels are propagated from template images to all other images that contain matching barcodes (gray arrows) and then transferred to the corresponding visible light images (red arrows). These labels are then used to train a deep learning model to predict landmark locations in novel images.

subside while the dye continues to fluoresce briefly (Fig. 2c, top). Note that longer decay processes are typically mediated by a mechanism known as phosphorescence, but we use the term fluorescence generically to refer to any decay process. This approach essentially eliminates both light leakage from the UV source and natural scene fluorescence (Fig. 2c, bottom). Moreover, because the UV source is not active during capture of UV images, there is no need for a low-pass filter to block UV light, resulting in a cheaper and simpler optical setup. The triphasic illumination approach enabled us to capture very large image sets with high signal-to-noise and without the need for manual removal of images with natural scene fluorescence. All experiments that follow were performed using triphasic illumination.

In image post-processing, we then compute label positions from the raw UV images (Supp. Fig. 1). First, the dye must be identified in the UV image. Owing to the very low background noise produced by the triphasic illumination scheme, we found that simple thresholding is sufficient to produce an accurate binary mask delineating the dyed region. Next, the binary mask is cleaned using morphological operators[29], which we use to smooth edges and eliminate small holes and blobs. Finally, the centroid (i.e., the center-of-mass) of the dye region is computed and stored as the label for the subsequent visible frame. For all frames in which the dye mask is empty after the cleaning step (e.g., scenes in which the landmark is occluded), the landmark position is recorded as absent and retained in the training set, as negative examples are important for effective model performance.

Even at high frame rates, on the rare occasions in which movements are very rapid, there can be a small discrepancy between the position of the generated label and the true landmark position on the subsequent frame. To determine if these cases present a problem, we used linear as well as bicubic interpolation to fill in the landmark position for each visible frame based on neighboring UV frames. However, we found that this interpolation scheme generally did not improve the final performance metrics (see model evaluation below), and therefore it was not used in the experiments that follow.

After synthesizing a labeled set of images, the final step in the serial labeling pipeline is to train a deep-learning model to perform landmark detection. For the experiments reported here, we specifically used the heatmaps and refinement maps of the DeeperCut model[30], but other architectures could also be used. One issue that must be addressed is how to treat absent landmarks. Depending on the model

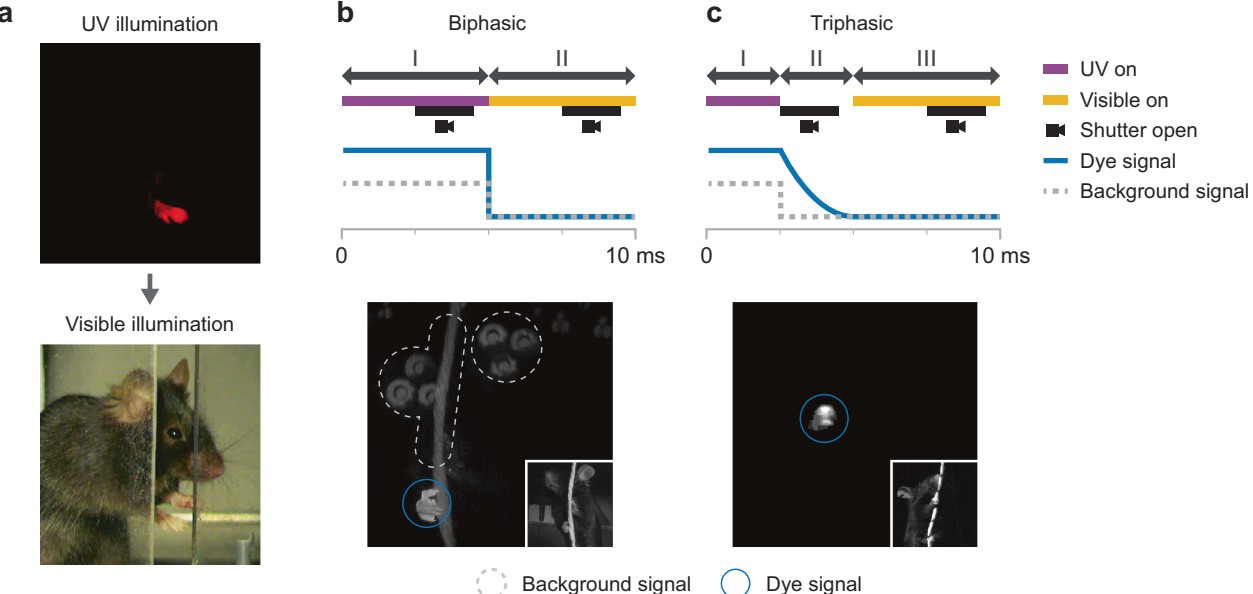

**Fig. 2 | Triphasic illumination with delayed-decay fluorescent dye eliminates background signals. a** Adjacent video frames captured under UV illumination (top) and visible illumination (bottom). **b** Biphasic illumination and image capture scheme (top), in which the UV image is captured while the scene is UV-illuminated. Example UV image from the biphasic scheme (bottom) showing two major sources of background signal (fluorescence of background objects, and broadband light from the UV source; dashed outlines) that make the dye region (blue circle) difficult to isolate, especially in monochrome images. **c** Triphasic illumination and image capture scheme (top), in which the UV image is captured after UV illumination has been turned off. A dye with a longer temporal decay constant is required. Example UV image from the triphasic scheme (bottom) showing that background signal is eliminated. Insets in **b** and **c** show corresponding visible illumination images.

architecture and the desired prediction behavior, different approaches could apply. The DeeperCut model produces a confidence map for each landmark. To account for absent landmarks, we configured the training procedure to encourage the model to produce an all-zeros confidence map for images in which the landmark was absent. The confidence map thus serves the dual purpose of encoding positional uncertainty and the probability that a landmark is present at all. To produce a final set of predictions from the model, we consider any predicted labels with confidence below a given threshold to represent landmarks that are absent (our evaluation metrics below do not require a fixed threshold, but rather measure performance over all possible thresholds).

**Increasing image diversity**
Fluorescence-based labeling can generate hundreds of labeled images per second, but those images tend to be highly redundant; the pose of the target does not change dramatically from one frame to the next, and the camera angle and lighting conditions do not change at all. In preliminary experiments we found that deep learning models trained on these images only produce accurate predictions in a limited range of visual environments.

To train a model with more visual diversity in the training data, we built a hemispherical dome with eight cameras and nine independently controlled light sources (Fig. 3a–c and Supp. Movie 2). The dome was assembled from custom 3D-printed parts and some off-the-shelf components (see Methods). Half of the cameras were monochrome and half were color, while four of the nine light sources were red and the others were white. Crucially, we designed custom LED holders along the periphery of hexagonal panels to ensure that UV illumination was omnidirectional and isotropic, avoiding UV shadows during video capture that would cause the dye to fail to fluoresce in some frames (Fig. 3b). Using a dome rather than a rectilinear system also allowed us to position the illumination sources at a uniform distance from the target, further reducing UV anisotropy.

By activating only one directional light source at a time during each successive frame, we produced images that exhibited a richer variety of lighting conditions (Fig. 3d, e). In addition, by capturing video from multiple cameras, we both increase the angular diversity of the data and collect more images per unit time (Fig. 3f and Supp. Fig. 2). Finally, animals performed several different behaviors on a circular platform that could be rotated throughout image capture, further contributing to image diversity (Fig. 3c and Supp. Fig. 2).

**Evaluating manual versus fluorescence-based labeling**
A key question we sought to answer was whether a large, diverse training set derived from hidden fluorescent labels could be used to train deep learning models that are more robust to varying visual contexts than the standard manual labeling pipeline, which often yields models that are hyper-specialized to a particular setting. To that end, we collected a large dataset of mouse behavior using the fluorescence pipeline described above to label one hand on the mouse. The full dataset contained ~380,000 images (388,496) representing eight camera angles, nine lighting conditions, and three behaviors: reaching for a pellet (captured at 100 UV/visible image pairs per second, with 5 ms delay within each pair), pulling a string (also 100 image pairs per second), and moving freely in an enclosure (2 image pairs per second) (Supp. Fig. 2). This dataset was divided into ~300,000 training images (299,736) and ~80,000 held-out test images (88,760). Note that unlike most evaluations in prior work, our training and test images are drawn from non-overlapping video clips, increasing the degree of difficulty and more accurately reflecting the challenge of domain generalization.

To emulate the kind of models commonly trained using the manual labeling workflow, we created subsets of the training and test sets respectively, both containing imagery from only a single camera and single lighting condition. We call these subsets uniform because they reflect a single experimental setup in fixed imaging conditions. Likewise, we call the full datasets, with their variety of camera angles and lighting conditions, the diverse sets. We trained three models with 250, 500, and 1000 images each sampled from the uniform training set. These quantities are commensurate with those typically used for manual labeling[11], and we refer to these models as having been trained

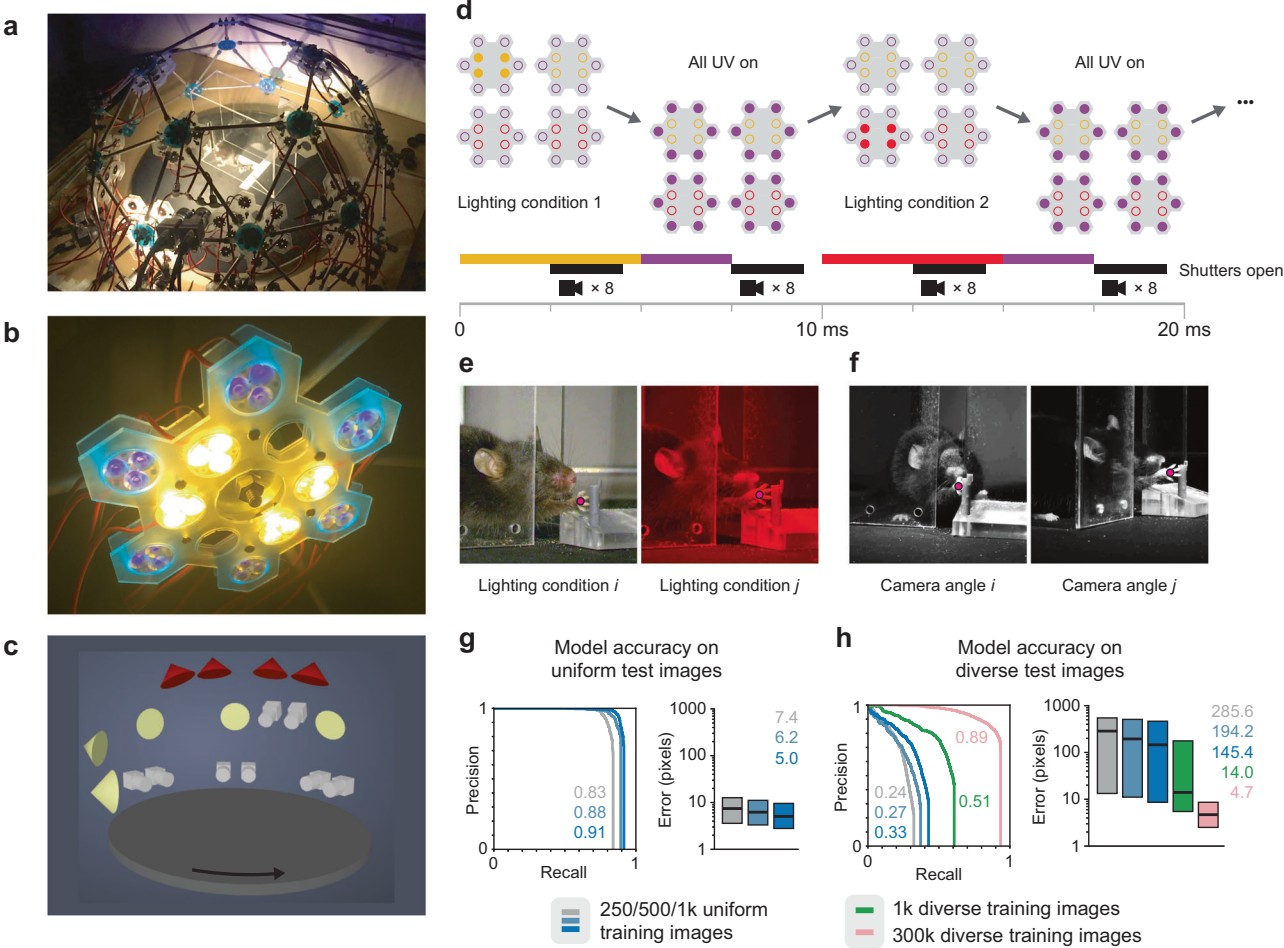

**Fig. 3 | A large, diverse training set derived from fluorescence imagery yields improved performance. a** A custom-made geodesic dome with eight cameras and nine LED clusters. **b** Each LED cluster contains visible LEDs at the center and UV LEDs at the periphery. **c** The spatial configuration of lights (5 white and 4 red) and cameras (4 monochrome and 4 color). Each light module is activated independently to produce images with different illumination angles. The platform inside the dome rotates to generate diverse azimuthal angles. **d** Sequence of illumination and camera triggers during video capture. All UV LEDs are triggered simultaneously to eliminate UV shadows. UV and visible illumination are activated alternately. During the visible illumination phase only a single cluster (white or red light) is active, the active cluster cycling sequentially. All 8 cameras capture simultaneously. **e** Two images captured under different lighting conditions on adjacent frames. Landmark labels (magenta dots) are derived from the corresponding fluorescence images (not shown). **f** Two images captured simultaneously from different cameras, increasing angular diversity in the training data. **g** Precision-recall curves (left; numbers indicate area under the curve) and pixel error quartile plots (right; center lines and numbers indicate median, boxes indicate 25th and 75th percentiles) of models trained on a single lighting condition and camera angle (uniform), and tested under the same conditions ($n = 1231$ test images). High accuracy is achieved with only a modest number of training images (250–1000). **h** When tested under diverse lighting conditions and camera angles, the accuracy of models trained on the uniform image set decreases significantly (blue and gray; $n = 4438$ test images). Training on a diverse (multiple cameras and lighting conditions) but small set of images (green, 1000 training images) shows some increase in accuracy but does not fully rescue performance ($n = 4438$ test images). Training on ~300 k diverse images results in significant improvements (pink) that are comparable to results in **g** ($n = 4438$ test images). As in **g**, center lines and numbers indicate median and boxes indicate 25th and 75th percentiles. Source data are provided as a Source Data file.

in the small-scale regime. The uniform dataset had an approximate image scale of 100–110 pixels per centimeter, and the diverse dataset had an approximate image scale of 50–110 pixels per centimeter.

To quantify model performance, we compared landmark predictions produced by the models to ground truth landmark locations using several metrics. Pixel error is defined as the distance between the predicted landmark location and its corresponding ground truth location. An on-target prediction is a prediction satisfying three conditions: the corresponding ground truth label is not absent (i.e., the landmark is visible), the prediction confidence is above a certain threshold, and the pixel error is below a certain threshold (5% of image width). Recall is defined as the number of on-target predictions divided by the total number of images in which the landmark is visible. Precision is defined as the number of on-target predictions divided by the total number of predictions with confidence above threshold.

Precision-recall (PR) curves are the set of 2D points traced out by computing precision and recall for all possible confidence thresholds. Area under the curve (AUC) is the integral of a PR curve; its maximum value is 1, with larger values indicating better accuracy.

In agreement with previously published results[11], we found that models trained on images from the uniform set achieved high accuracy when tested on held-out test images also drawn from the uniform set, even when the number of training images was less than 1000 (i.e., in the small-scale regime) (Fig. 3g). For instance, the model trained on 1000 images achieved an AUC of 0.91 and a median pixel error of 5.0. Next, to assess the robustness of these models to varying conditions, we tested them on held-out test images from the diverse set. We found that accuracy decreased substantially. For the model trained on 1000 images, AUC dropped from 0.91 to 0.33 and median pixel error increased from 5.0 to 145.4 (Fig. 3h). These results support the idea

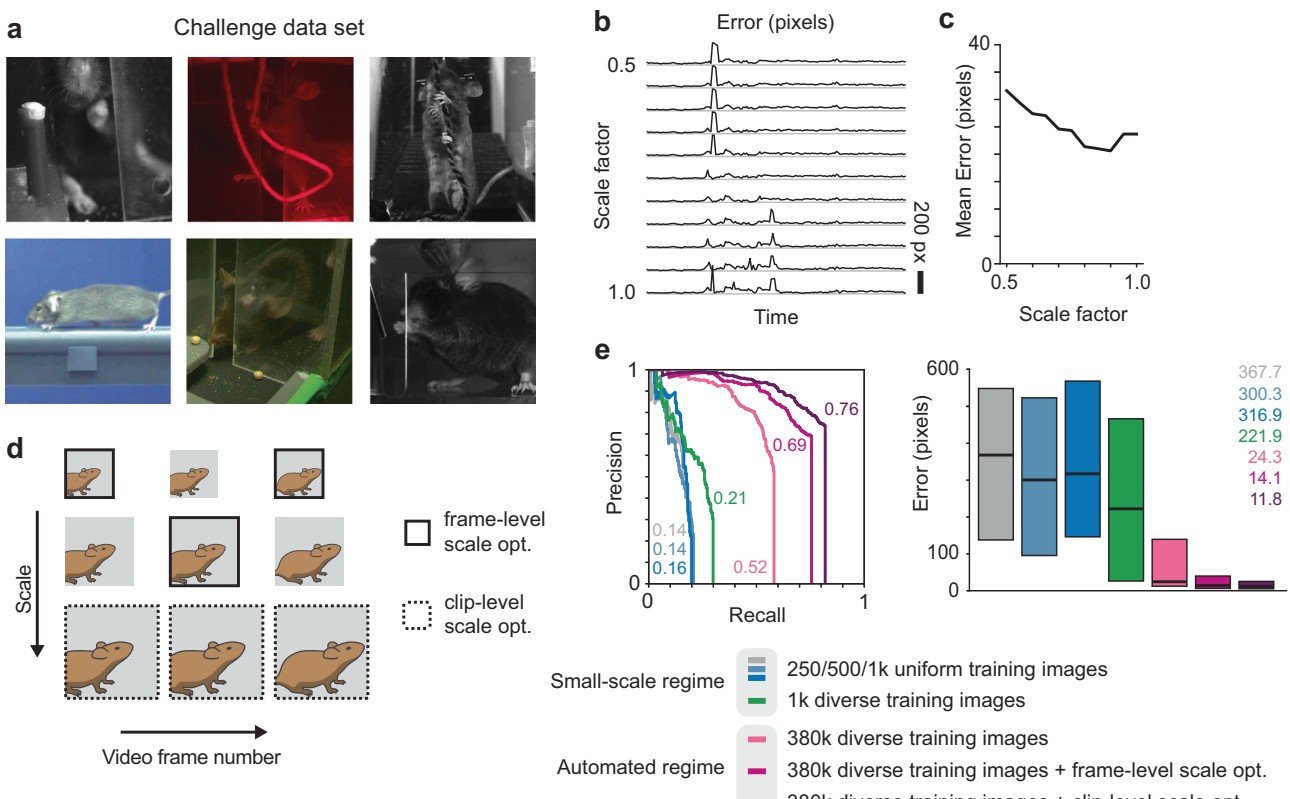

**Fig. 4 | Landmark detection on novel setups using test-time scale optimization.** **a** Sample images from a challenge dataset, collected from various experimental setups. **b** Pixel error when running a trained model on a video clip from the challenge set. Each row represents a different spatial scale factor applied to the video clip prior to landmark detection. **c** Mean prediction error in pixels versus scale factor. The model exhibits a preferred scale. **d** Two schemes for test-time scale optimization: frame-level (solid) and clip-level (dashed). In frame-level optimization, each frame receives its own scale factor. In clip-level, all frames share the same scale factor. **e** Precision-recall curves (left; numbers indicate area under the curve) and pixel error quartile plots (right; center lines and numbers indicate median, boxes indicate 25th and 75th percentiles) for the different types of scale optimization compared to performance with no scale optimization as well as small-scale regime networks from Fig. 3g, h ($n = 612$ test images). Clip-level scale optimization achieves the highest performance. Source data are provided as a Source Data file.

that models trained in the small-scale regime are sensitive to imaging conditions.

A natural question is whether this sensitivity to conditions is due to the small size of the training set or its lack of visual diversity. Therefore, for comparison, we also trained a model with 1000 images drawn from the diverse set and evaluated this model on the diverse test set (Fig. 3h). While this model achieved better accuracy (AUC = 0.51, median pixel error = 14.0), these results still represent a significant drop in performance compared to the models trained and evaluated on the uniform dataset. This result suggests that increasing training image diversity is not sufficient to rescue performance on a diverse test set; for a model to perform well under variable conditions, more training data is required. To test this idea, we trained a model on a large subset of the diverse set containing all 300,000 images that were not part of the held-out test set (Fig. 3h). When evaluated on images drawn from the diverse test set, this model was far more accurate (AUC = 0.89, median pixel error = 4.7), comparable to performance when uniform training and test data are used (Fig. 3g).

Evaluating model performance on a set of test images requires ground truth labels to serve as a basis of comparison for the labels produced by the model. In all the above evaluations, we used fluorescence-derived labels as ground truth. To confirm that fluorescence-derived labels are indeed a valid source of ground truth, we manually labeled a subset of the diverse test set and repeated the same evaluation with these manual labels serving as ground truth

(Supp. Fig. 3a). The results did not change, confirming that fluorescence-derived labels can serve as valid ground truth for evaluation. Finally, we evaluated all models using the object keypoint similarity mean average precision (OKS-mAP) metric, another standard performance measure that takes into account object scale and annotation uncertainty[31], revealing similar results (Supp. Table 1). Together, these findings indicate that training on a large-scale dataset generated using our automated fluorescence labeling technique can achieve robustness to variable imaging conditions that is not achieved by models trained on smaller and less diverse datasets typical of setup-specific manual labeling.

## Robustness across experimental setups

While the diverse set contains a wide variety of imaging conditions, the true heterogeneity seen across behavioral setups and laboratories is not sufficiently represented. To assess the versatility of this labeling approach more rigorously, we next examined the extent to which a model trained on a large-scale fluorescence dataset could generalize to visual contexts completely different from the one it was trained on. We collected a challenge dataset from archival video (monochrome and color) of mice performing five different behaviors across ten different visual environments across two laboratories, with large variety in animal scale (approximate image scale of 25–160 pixels per centimeter) and resolution (Fig. 4a). To acquire ground truth labels for evaluating prediction accuracy, a subset of frames from the challenge set were manually labeled by human annotators.

When evaluated on the challenge set, the models trained in the small-scale regime (Fig. 3g, h) performed poorly (e.g., a model trained on 1000 images from the diverse set produced AUC = 0.21, median pixel error = 221.9). In contrast, training with the full set of over 380,000 images from the diverse set resulted in a boost in accuracy, doubling AUC and reducing pixel error by a factor of 9 (AUC = 0.52, median pixel error = 24.3) (Fig. 4e). Despite this increase in accuracy by increasing the quantity of training data, we sought to further improve performance. Given the wide variability in image scale across the challenge set, we hypothesized model accuracy could be improved by rescaling the dimensions of each test image prior to processing by the trained model, thus changing the apparent size of the animal to more closely match that seen in the training data. Testing this idea, we found that applying a range of different pre-scaling factors (0.5 to 1.0) to the frames of a test clip before feeding them to the deep learning model resulted in different predicted hand trajectories of varying accuracy (Fig. 4b, c). The fact that model predictions tend to be most accurate for a particular, but unknown, pre-scaling factor that varied by video clip motivated us to implement a process of scale optimization at test time to improve accuracy.

The goal of scale optimization is to select the image pre-scaling factor that causes the model to produce predictions that exhibit minimum pixel error. However, in the general case, ground truth labels are not available at the time the model is applied to a video clip, since it is precisely when labels are missing that the model is needed; therefore, pixel error cannot be computed directly. To address this issue, we decided to optimize confidence values as a proxy metric instead. Scale optimization with the confidence metric can be applied independently to each frame (frame-level scale optimization), or it can be applied uniformly to an entire video clip such that all frames share a single scale factor (clip-level scale optimization) (Fig. 4d). Both styles of scale optimization yielded significant accuracy improvements over the raw model output. A model trained on the full diverse set (380,000 images) and tested with frame-level scale optimization achieved an AUC of 0.69 and a median pixel error of 14.1, while clip-level scale optimization achieved an AUC of 0.76 and a median pixel error of 11.8, well above performance with no scale optimization applied (Fig. 4e and Supp. Movie 3). As above, the OKS-mAP metric produced similar results (Supp. Table 1). It was somewhat surprising that clip-level scale optimization consistently produced more accurate results than frame-level optimization. We speculate that this may be due to the fact that maximizing confidence rather than minimizing true error introduces some noise into the value of the optimal scale, while averaging confidence over all frames tends to reduce the magnitude of this noise term.

To further explore the impact of training set characteristics, we performed a series of ablation experiments, reducing the number of camera angles, lighting conditions, and overall training dataset size, each independently. We found that modifying any of these parameters impacts performance, with training dataset size having the largest impact (Supp. Fig. 3b). We also note that several established temporal smoothing approaches are compatible with the output data generated by these trained models and could be used to further reduce anomalous labels[32]. Moreover, systems that perform 3D pose estimation with multiple cameras can further reduce errors by identifying when a detection from one camera disagrees with the detections from the other cameras[32].

Finally, to verify that the serial labeling approach is capable of training multi-landmark models, we developed a training method for combining multiple datasets, each containing a different labeled landmark, to train a single model. The typical approach to training a multi-landmark network is to label all landmarks in each frame (Supp. Fig. 4a). We implemented another method, which we call interleaved training, in which only the landmarks that are labeled in a particular training image contribute to the loss function (Supp. Fig. 4b). We used

fluorescence labeling to collect an additional training dataset (126,920 images) in which the mouse foot was labeled. When tested on the challenge set, a combined, multi-landmark model trained on both the mouse hand and foot performed similarly to two separate models trained on the hand and foot respectively, showing that the interleaved training scheme can be used to build up multi-landmark models from single-landmark datasets (Supp. Fig. 4c).

Together, these results demonstrate that large, diverse datasets are needed to train models capable of generalizing to different experimental setups, and that our fluorescence labeling approach can provide the necessary quality and quantity of training data.

## Interactive optimization using live feedback

Even a deep learning model with good robustness can achieve higher accuracy when presented with input images that are more like the data it was trained on. Therefore, after a model is trained, another way to improve performance is to adjust the scene when collecting new video data to make the images captured by the cameras more suited to the preferred image properties of the model. We tested this optimization idea by designing a user interface for displaying the predictions of a deep learning model in real-time (Supp. Fig. 5a). The interface displays the most recent camera frame overlaid with the model's landmark prediction. It also displays a line plot of the x-position, y-position, and confidence of the most recent $k$ landmark predictions (for our experiments, $k$ was set to a value of 100, and average neural network throughput was approximately 10 frames per second).

To evaluate the usefulness of real-time feedback for increasing performance, we set up a camera to capture a head-fixed mouse performing a water-reaching task (a behavioral setup not seen by the deep learning model during training). We collected a few seconds of video under three different conditions while monitoring model output. Between each video recording, the camera position and aperture settings were manually adjusted using real-time feedback to increase confidence and the stability of predictions. After video capture, we manually labeled these video frames for ground truth and evaluated model performance in each of the three clips (Supp. Fig. 5b). Performance improved in each of the successive camera adjustments, supporting the notion that optimizing a behavioral setup to suit a particular deep learning model using real-time kinematic and confidence value feedback is a feasible way to further improve performance during data collection.

## Visual barcodes for massively parallel labeling

While the serial labeling pipeline can be repeated for each landmark we wish to track, this repetition becomes less practical if one wishes to track a large number of landmarks. To address this challenge, we adapted the fluorescence strategy to develop a parallel labeling approach that drastically increases the number of landmarks that can be labeled simultaneously. The fundamental idea is that local regions receive distinctive visual barcodes that can be tracked in parallel, yielding many landmark labels per frame instead of just one.

Using the human hand as proof-of-concept, we tested several methods for generating random visual barcodes: applying fluorescent powder suspended in transparent adhesive, applying liquid dye with an airbrush, and applying liquid dye by agitating the bristles of a brush to produce a fine aerosol. We found that the brush aerosol approach was most effective because of the scale and uniformity of the speckle pattern it produced (Fig. 5a). From the perspective of data collection, the only difference between serial and parallel labeling is the method by which the dye is applied: via aerosolization rather than with a marker. To validate our parallel labeling approach, we collected a video dataset containing 12,276 frames of a speckled hand captured at a rate of 10 image pairs per second, using the same dye, lighting, and cameras as above (though only the monochrome cameras were active). Finally, we captured 500 frames of video of an unlabeled hand

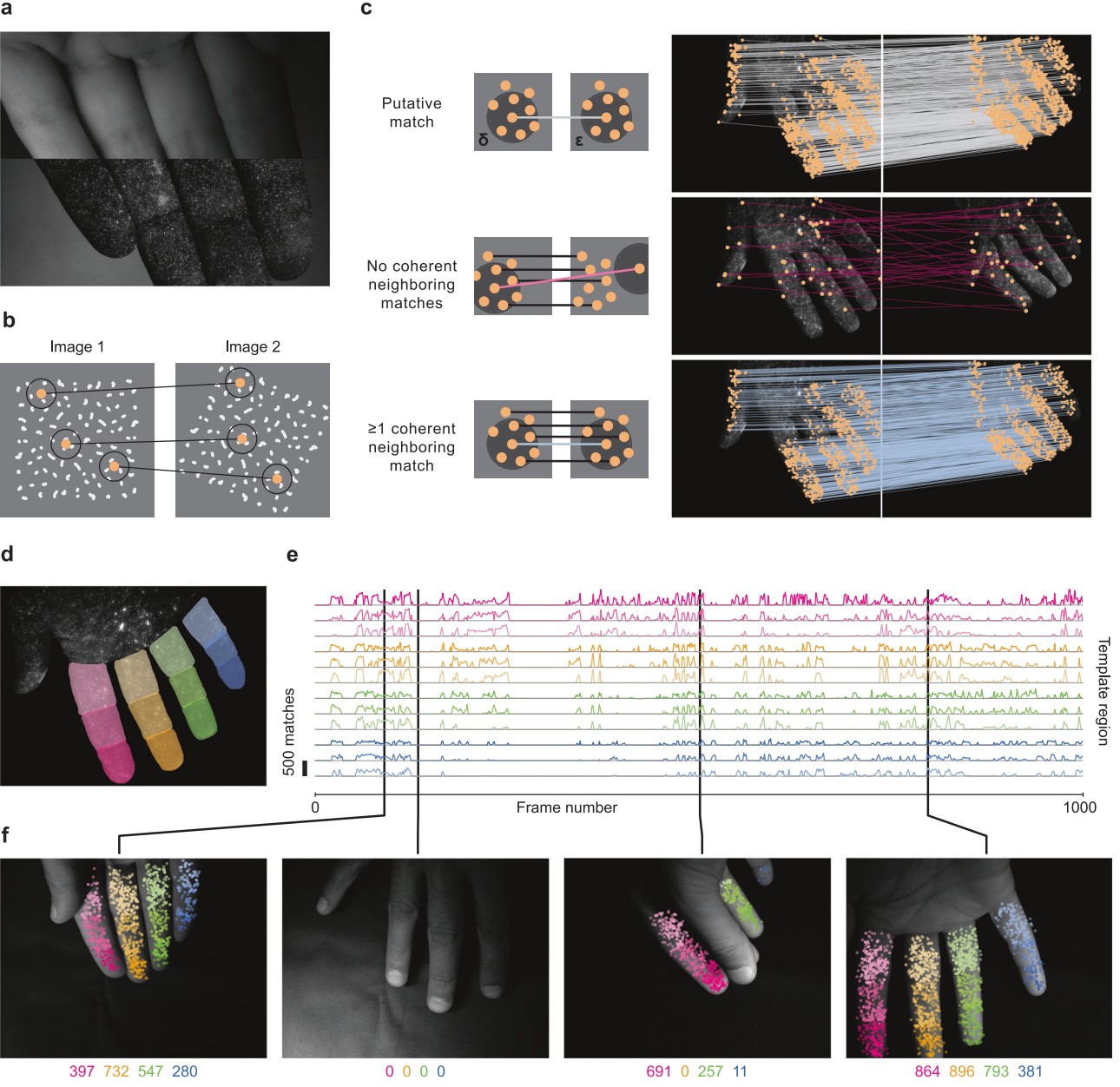

**Fig. 5 | Visual barcoding for massively parallel labeling. a** Adjacent frames of visible illumination (top) and UV illumination (bottom) of the hand with hidden fluorescent speckle pattern. **b** Matching of visual barcodes between images using the scale-invariant feature transform (SIFT) algorithm. Each point represents a visual barcode, defined as a SIFT feature. The circle around each point represents the local image region from which the barcode descriptor is computed. **c** A putative match (top left, gray line) between two barcodes in a pair of non-consecutive images. The dark disks around each barcode represent circles of radius δ and ε. All putative matches between barcodes in two images of the hand (top right). A putative match (middle left; pink line) with no coherent neighboring matches (black lines) is likely to be an incorrect match. All such matches between the two

hand images (middle right), showing that these matches are nearly all spurious. A putative match (bottom left; light blue line) with at least one coherent neighboring match (black lines) is likely correct. All such matches between two hand images (bottom right). **d** A template image with twelve manually labeled regions delineating twelve finger segments with the palm facing up. **e** Twelve curves representing the number of SIFT matches between each video frame and the twelve finger segments (sum over ten labeled template images; see text). **f** Four sample frames showing SIFT matches corresponding to the frame in **e**. Numbers indicate sum of SIFT matches on each digit (three finger segments). When the palm is not visible (image 2) or a digit is occluded (image 3), no matches are detected. Source data are provided as a Source Data file.

with a different camera and lighting setup to be used as test data (the image scale was approximately 100 pixels per centimeter).

Visual barcodes are only useful for training deep learning models if they can be matched across images. For example, if the tip of the finger rotates in front of the camera as the hand moves, the barcodes from one frame need to be matched to the same barcodes in the other frames despite changes in orientation and appearance. To address this challenge, we applied scale-invariant feature transform (SIFT)[33], a computer vision technique for matching distinctive keypoints (also

known as features) between images (Fig. 5b). While other feature-matching algorithms that have been developed in recent years could also be used effectively, we selected the SIFT approach as it performs well on several quantitative benchmarks[34,35]. We found that matching SIFT features between pairs of UV images produced hundreds to thousands of matches, a small minority of them (typically <10%) being spurious (Fig. 5c).

To remove incorrect matches, we developed a spatial coherency heuristic as a filter. Considering a putative match, we defined a circle of

                                                    

radius δ around its left endpoint and a circle of radius ε around its right endpoint (Fig. 5c, top left). For the match to be valid, it should have at least one coherent neighboring match: a match whose left endpoint is within the left circle and whose right endpoint is within the right circle. For the experiments shown here, we set δ and ε to 50 pixels. Visual inspection shows that most matches with no coherent neighboring matches are spurious (Fig. 5c, middle), and that the far more abundant matches with at least one coherent neighboring match are correct (Fig. 5c, bottom).

To visualize the number and accuracy of SIFT matches, we found it convenient to color code them according to which part of a template image they match. We accomplished this by manually labeling template images with an arbitrary number of different regions, in this case the twelve segments of the digits with the palm facing up (Fig. 5d). Denser SIFT coverage can be accomplished by selecting several template images, given the wide variety of configurations the hand can make. Moreover, while we focus on the palm surface of the hand, any hand region can be prioritized based on which template images are selected. To help select a set of template images that achieve good matching coverage and are not too redundant (i.e., templates containing a diversity of poses with the 12 digit segments mostly visible), we devised a greedy algorithm that iteratively selects as a template the next image estimated to match with the greatest number of unmatched SIFT features (Supp. Fig. 6), which we then supplement with a small amount of human filtering. With this visualization approach, we find dense coverage of SIFT matches across all twelve digits, with essentially no spurious labels on the incorrect digit segment or when the segment is occluded (features pooled over 10 templates; 1000-frame clip) (Fig. 5e, f and Supp. Movie 4). These results support the idea that thousands of fluorescent barcodes can effectively be matched across images. Moreover, only a small number of template frames (10 in this case) are needed to effectively cover the full 12,000-frame dataset, demonstrating the strong scaling properties of the approach.

In addition to using the fluorescent speckle pattern to produce labels, it can also be used to segment the object of interest from the background (Supp. Fig. 7). With accurate segmentation, the training set can be augmented by replacing the background with different random images. In our experiments, the random images were generated from a complex noise distribution, but they could alternatively be sampled from a set of natural images. This type of augmentation improves the accuracy and robustness of the resulting landmark detection models and was used for training the models reported below.

**Using visual barcode matches to train deep learning models**
Given a set of SIFT matches, additional processing is necessary to produce landmark labels that can be used as training data for deep learning models. For example, in a simple scheme we selected raw SIFT features from a template frame to act as landmarks. We then generated landmark labels for all other frames by simply matching the template SIFT features to the SIFT features in every other frame. Features that matched successfully received an (x, y) label, while those that did not were marked as absent. The downside to this scheme is that every failure in SIFT matching is translated directly into an erroneous training label. While SIFT matching with spatial coherency filtering produces very few false positives, there remain a problematic number of false negatives, as individual SIFT matches often drop out frame-to-frame. We found that deep learning models trained on landmarks derived directly from raw SIFT features failed to produce reasonable predictions (i.e., they had low correlation with ground truth positions), possibly indicating that training failed to converge to good model parameters due to the large number of erroneously missing labels. To address this issue and reduce labeling errors, we developed two schemes for pooling SIFT features together into local neighborhoods

to provide resiliency to failed matches. The first scheme, which we call manual neighborhood selection (Fig. 6a–f), involves a modest amount of manual annotation; the second, which we call automatic neighborhood selection (Fig. 6g–l), is fully automated.

In manual neighborhood selection, the user selects a template image from the training set and manually draws a set of neighborhoods of interest over it, each representing a single landmark to track (Fig. 6a, b). The SIFT features within each segment then act as pooled template features for one landmark. These neighborhoods can be any size the user decides, but for proof-of concept we used the same digit segment templates used above for visualization (Fig. 5d–f). We then had these pooled features 'vote' on the location of the landmark in each training image by computing SIFT matches and then computing the centroid of all successful matches. In this way, the impact of false negatives is greatly reduced; as long as a few true positives are present, the centroid of the segment will be estimated accurately.

To further increase the reliability of matching, we labeled finger segments in not just one but ten template images, as described above (Supp. Fig. 6). We found that using more templates produced more matches, and that the greedy template selection algorithm resulted in 3 to 5 times as many SIFT matches as compared to random template sampling (Fig. 6c). We then produced labeled training images with centroids overlaid onto the visible image of the hand across the image dataset (Fig. 6d). For some images, no single template image was able to provide SIFT matches for all 12 segments, reinforcing the usefulness of using multiple optimized template images (Supp. Fig. 8).

We next used the labeled training data to train a network to identify our manually delineated landmarks (the 12 finger segments in this case) on unlabeled test data. To measure the impact of the number of template images on end-to-end accuracy, we trained three deep-learning models with labels generated from one, five, and ten template images. These models were evaluated on manually labeled images from a test set, and, as expected, the larger number of template images improved accuracy significantly (e.g., ten templates: AUC = 0.76, median pixel error = 36.5; one template: AUC = 0.37, median pixel error = 72.7) (Fig. 6e, f and Supp. Movie 5).

Labeling the ten template images required about 30 min of labor. While this approach is feasible for a modest number of landmark neighborhoods (12 in this case), the amount of manual labor scales linearly with the number of landmarks and becomes more difficult with smaller neighborhoods that are harder to distinguish. We, therefore, sought to develop a fully automated method that could be scaled up to a practically arbitrary number and density of landmarks limited only by the underlying SIFT matches.

With automatic neighborhood selection, the local neighborhoods over which SIFT matches are pooled are circles of a user defined radius around each landmark feature (Fig. 6g). In this case, landmark features were selected on a single digit to be at least 30 pixels apart, giving relatively uniform coverage of the surface (Fig. 6h). Landmark labels were computed as follows: if a landmark feature matches a particular training image, then the match location is used as the landmark label. However, if a landmark feature fails to match but at least k other features in its local neighborhood match successfully, then the neighborhood matches are used to fit a local homography and project the template feature into the target image[36,37]; the projected location is then used as the landmark label (Fig. 6i, left). The homography scheme greatly increases the percentage of landmark features that are successfully transferred to the target image, from less than 10% to over 50% (Fig. 6i, right; k = 10). Note that the maximum percentage is less than 100% because landmarks are absent in some frames. Training images with landmark labels were then generated across the entire dataset (Fig. 6j).

We next trained a deep learning model on 50 landmarks from a single digit generated with automatic neighborhood selection. Evaluation of network performance using automatically selected

                                                                                     

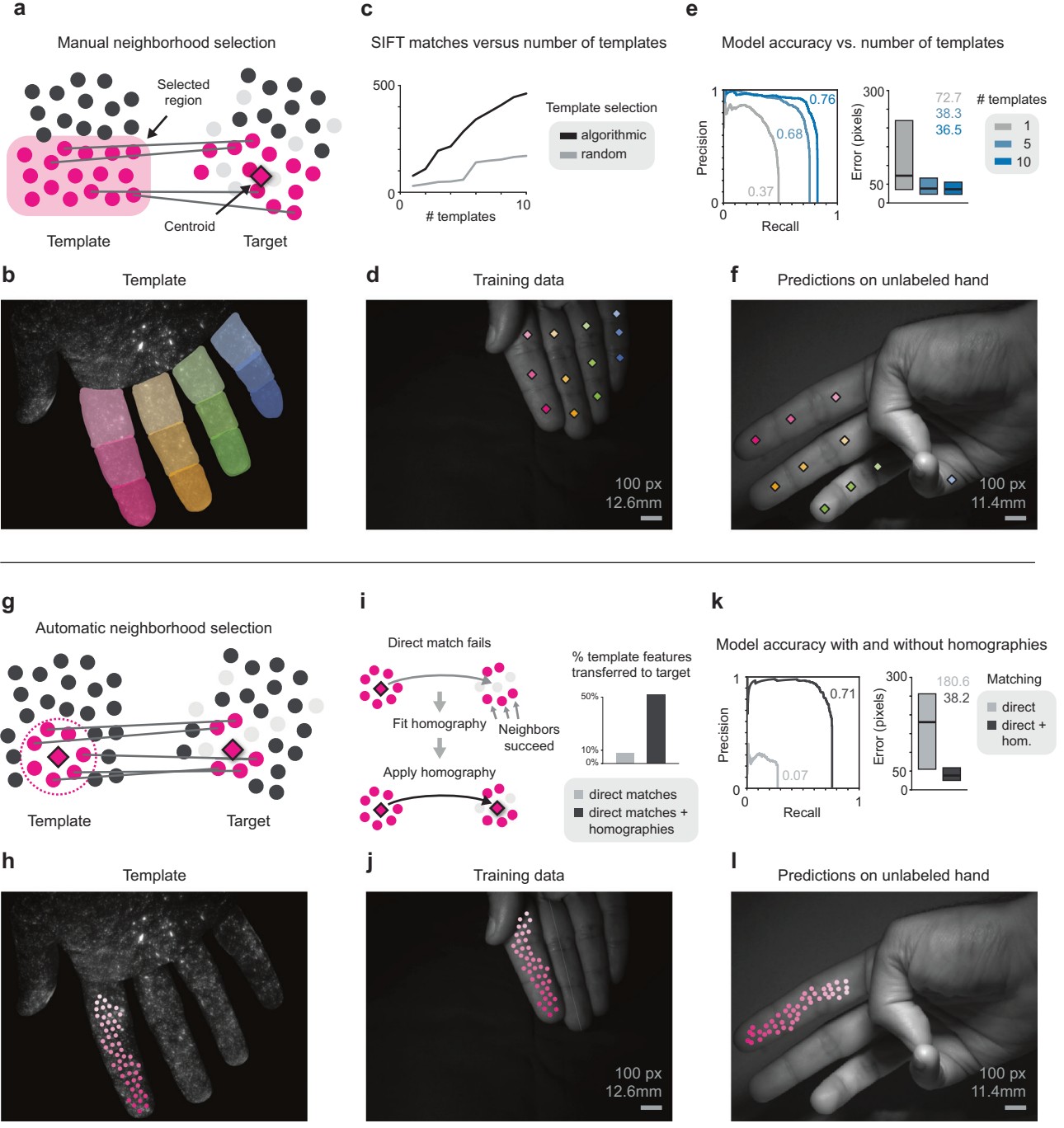

**Fig. 6 | Pipelines for training landmark detectors from visual barcode data. a–f** Manual neighborhood selection. **a** A template image (left) is manually annotated with neighborhoods (light pink). Scale-invariant feature transform (SIFT) features in the template image (pink circles) are matched to SIFT features in a target image (right). Successful matches from each template neighborhood are pooled to produce a single landmark label (centroid) in the target image (diamond). **b** Example template image with twelve neighborhoods. **c** Greedy algorithmic selection (black) produces more SIFT matches from a given number of template images compared to random selection (gray). SIFT matches increase with number of templates used (see Supp. Fig. 6). **d** Example training image with labels derived from ten annotated template images transferred to the visible light image. **e** Precision-recall curves (left; numbers indicate area under the curve) and pixel error quartile plots (right; center lines and numbers indicate median, boxes indicate 25th and 75th percentiles) from testing on held-out data ($n = 2400$ test labels). Using more template images increases performance. **f** Held-out evaluation image labeled by the trained model. Occluded regions do not receive a label. **g–l** Automatic neighborhood selection. **g** SIFT features are grouped by proximity to an active feature (left, diamond). **h** A template image with features selected via spatially uniform subsampling on the first digit. The minimum distance between selected features is 30 pixels. **i** If the active feature fails to match the target image, the neighborhood matches within a defined radius are used to estimate a local homography and reproject the active feature from template to target (left). The number of template features transferred to the target increases if homography is used. **j** Example training image labeled using local homographies. **k** Precision-recall curves (left; numbers indicate area under the curve) and pixel error quartile plots (right; center lines and numbers indicate median, boxes indicate 25th and 75th percentiles) from testing on held-out data ($n = 2400$ test labels). Using local homographies increases performance. **l** Held-out evaluation image labeled by a model trained with homographies. Source data are provided as a Source Data file.

landmarks is challenging because of the large number of landmarks and because a human annotator cannot easily label arbitrary surface landmarks for ground truth (though, with the proper computational infrastructure this is possible[38]). We therefore evaluated the models against the manual labels of the digit segments used to evaluate manual neighborhood selection above. One landmark feature was selected as a proxy for each digit segment: specifically, the landmark feature that was closest to the ground truth label for that digit segment averaged over the entire evaluation set. We found that the use of homographies improved accuracy significantly when compared to a model trained on data labeled only from direct feature matches (AUC = 0.71 vs. 0.07, median pixel error = 38.2 vs. 180.6) (Fig. 6k). Using this approach, our model could reliably identify dozens of landmarks on an unlabeled hand (Fig. 6k, l and Supp. Movie 6). More generally, these results show that rather than tracking a small number of visually identifiable points on a complex object like the hand, as is feasible with manual landmark annotation, one could approach dense coverage of the skin to monitor the contours and deformations as the object moves through space. A complete overview of the parallel labeling workflow is included in the supplemental materials (Supp. Fig. 9).

## Discussion

We have described an approach for generating training datasets of sufficient size and diversity to create more versatile deep learning-based motion capture models. This large increase in volume and quality of training data is complementary to and can be combined with the many advances in training procedures and model architectures that are ongoing in the field of deep learning. Our methods can be used to label individual landmarks of interest serially, or large numbers of landmarks in parallel. Together, the scalability of these approaches enables automated markerless tracking with less effort and greater generalizability to new visual environments than is typically feasible with manual annotation approaches.

The field of markerless pose tracking for scientific applications has seen a surge of new methods as software tools and technical ideas from the human pose detection literature, particularly related to deep learning, have been adapted and optimized for new purposes[11–14,32,39,40]. Up to now the dominant paradigm for pose tracking in the laboratory has been to train new deep learning models for each experimental setup or dataset, but the field is increasingly moving towards more versatile, generalizable models that can be used in different settings without re-training. This trend is evident in several recent advances that, like ours, focus on synthesizing large-scale datasets. Within the realm of rodent pose tracking, notable approaches for generating labeled training data include: DANNCE[18], which uses a multi-camera setup and 3D reconstruction to propagate labels to many cameras; CAPTURE[19], which uses markers implanted in the skin; and work that uses a virtual 3D rodent to generate synthetic data[20]. For monkey pose tracking, OpenMonkeyStudio[41] developed a large training set using a multi-camera setup and label propagation, while MacaquePose[42] labeled a large set of images of macaques in the wild. In the realm of human pose tracking, the HumanEva[5] and Human3.6M[43] pose datasets were created using marker-based motion capture, while the PoseStudio[44] dataset was created using a multi-camera setup, 3D reconstruction, and label propagation. The FreiHand dataset[28] for human hand pose used a deformable 3D model to extract finger poses from multi-camera imagery.

Here, we take a different approach to generating large datasets for training pose estimation models – fluorescence labeling. Compared to other label generation techniques, fluorescence labeling has the advantage that it does not require a 3D model nor 3D reconstruction (which can pose challenges for some subjects). The approach is also simple in terms of computational requirements, making it an attractive option for many labs. Although this is conceptually a marker-based technique, fluorescent dye need only be applied to one or a small cohort of subjects to generate very large amounts of training data, and it may be tolerated better by laboratory animals than retroreflective markers or opaque paint. Some caveats to consider are that the dye must be compatible with the subject and the illumination intensity for dye excitation must be scaled up in proportion to the square of the distance to the subject. The size of the dye region must also be large enough to be resolved consistently by the cameras.

There are several prior demonstrations of fluorescent imagery being used as a source of ground truth data for training predictive models. For instance, fluorescent labeling has recently been used to train deep learning models to predict features of interest in microscopy imagery[45,46]. Hidden fluorescent labels have also been used to generate ground truth data for computer vision, the most relevant for our approach being the Middlebury optical flow dataset[21]. In that work, several scenes were painted with a hidden fluorescent speckle pattern, and each scene was photographed under visible and UV illumination, both before and after a small perturbation to the scene. The speckle pattern visible in the UV images was used to compute the optical flow (i.e., the dense 2D correspondence map) between the images of the perturbed and unperturbed scenes, and ground truth data was used to optimize optical flow algorithms.

We extend the core idea of fluorescent labeling with several innovations: a) the use of fluorescent labels for training markerless tracking models; b) the use of triphasic illumination for elimination of background fluorescence; c) the use of multiple light sources, camera types, camera angles, and behaviors to mitigate visual redundancy and increase diversity; and d) the use of visual barcodes and SIFT features to track and label many landmarks in parallel. While our work focuses on motion capture for life sciences research, the fluorescence labeling approaches we developed are general and could be used for other pose tracking applications in which objects of interest can be imaged offline, in a controlled setting, prior to model training.

Our scale optimization procedure is related to a large body of literature on keypoint estimation and test-time augmentation, especially in humans[47–49]. Like our method, test-time augmentation involves applying a neural network to different transformed versions of a specific test image and processing the resulting predictions to yield a single higher-accuracy prediction on the original image. Much work in test-time augmentation has focused on the single-image case, where, unlike with our data, information is not shared across the frames of a video. To combine the multiple predictions produced for each image, various strategies have used either the mean[47,49] or a learned function[48]. Our work suggests that test-time augmentation across multiple video frames may be a promising future direction for investigation. At the same time, heuristics like ours that optimize confidence scores may also result in higher false positive rates if not applied judiciously. For instance, the confidence threshold, if used, will need to be adjusted appropriately.

Our parallel labeling approach is also related to much prior work in computer vision, including unsupervised keypoint detection, surface tracking, and pose-based supervision for landmark detection. Unsupervised keypoint detection is the process of discovering visual keypoints from image data without the experimenter providing specific labels. It has been shown, for instance, that a neural network can be trained to detect an arbitrary number of keypoints given many images of a rigid object from different views, using only relative camera viewpoints of the different images as a supervision signal[50]. Similarly, neural network autoencoders have been used to automatically extract potential landmarks from large image sets containing human faces, cat heads, and human poses[51]. Surface tracking is the task of tracking each point on a surface over time, typically from successive frames of a video-type data stream. Some notable examples include DynamicFusion[52], in which depth data is used to track a non-rigid object in real time, and deformable neural irradiance fields[53], where a

non-rigid object is tracked and reconstructed from a series of 2D images. Pose-based supervision for landmark detection involves using pose data from one data source to train a landmark detector operating on a different data source (typically monocular imagery). Some examples include training a monocular hand pose detector from depth data[54], and training a monocular human pose detector from multi-camera data with partial labels[44].

While the parallel labeling approach based on SIFT features offers a powerful way to densely track deformable surfaces like skin, it still suffers from being limited to surfaces with certain properties. In particular, adapting parallel labeling to track many landmarks on the mouse still faces challenges with fur and is a goal of future work. Another consideration is that the precise number of points trackable in parallel will depend on the density of the speckle pattern, which determines the spatial density of SIFT features. In addition, most commercially available fluorescent dyes decay too quickly to be used for our triphasic illumination scheme; we were only able to find one dye with the required millisecond-scale decay constant. Looking forward, the development of more dyes with millisecond-scale fluorescence lifetimes and a variety of emission wavelengths would enable a parallel fluorescence labeling approach based on color multiplexing that would combine some of the strengths of the serial and parallel labeling approaches presented here and would work well on a wider range of surface types, including on fur. Fluorescent and phosphorescent dyes for microscopy is one possible area relevant to developing more dyes with the desired properties[46,55–59].

With more versatile and automated pose estimation tools, behavioral research fields are generating rapidly growing amounts of movement data. Analytical techniques that can make sense of this type of motion data, several of which have already been developed[60–64], will continue to grow in usefulness. For example, techniques that can detect and characterize subtle changes in movement during early stages of disease could impact how prodromal pathology is detected and how therapeutic interventions are evaluated. Ultimately, these kinds of advances in tracking and quantifying movement can serve to support and augment the insights of well-trained human observers.

## Methods
Animal procedures performed in this study were conducted according to US National Institutes of Health guidelines for animal research and were approved by the Institutional Animal Care and Use Committee of The Salk Institute for Biological Studies. All human subjects work presented here complied with all relevant ethical regulations, was performed with informed consent, and was designated to meet the criteria for the Benign Behavioral Interventions exemption by the Institutional Review Board of The Salk Institute for Biological Studies.

### Hardware and software
Video capture was performed using eight USB3 cameras: four monochrome (Basler Ace acA1440-220um with 16 mm Tamron lens M118FM16) and four color (Basler Ace acA1920-155uc with 8 mm Tamron lens M118FM08). The cameras were triggered by an Arduino Due microcontroller over GPIO output pins. Image capture was performed with a desktop computer using the Basler Pylon 5 C + + API.

UV illumination was provided by thirty LED modules (400–410 nm; LED Supply, A008-UV400-65), each containing three LEDs, and mounted in five hexagonal clusters of six modules each. Each hexagonal cluster was constructed from a custom 3D-printed faceplate and backplate held together with machine screws, with the LED modules held in between. Visible illumination was provided by a similar LED configuration: five clusters containing four white LED modules each (LED Supply, CREEXHP35-765-3) and four clusters containing four red LED modules each (LED Supply, CREEXPE2-RED-3). Red LED modules were used because it is common in rodent behavioral experiments to shift the light/dark cycle and expose mice to only red light during behavioral experiments. LEDs were controlled by the same Arduino Due driving power MOSFETs, and power was supplied by a BK Precision 1672 variable power supply (Test Equipment Depot, 817050167207).

Cameras and LEDs were mounted to a 2 V geodesic dome. The dome was constructed out of ¼-inch aluminum cylindrical struts with 3D-printed mounting attachments bonded to each end using a two-part epoxy (J-B Weld 50176 KwikWeld). Each vertex of the dome was a 3D-printed circular hub to which the struts were fastened with screws. This design allowed individual parts of the dome to be removed and re-arranged as needed by unfastening the appropriate screws. All 3D printing was performed with a FormLabs Form 2 using FormLabs Tough Resin (RS-F2-TOTL-05).

Inside the dome, the animal's behavioral apparatus was placed on a rotary turntable consisting of a ½-inch acrylic disk on a circular track. The acrylic disk was covered in a ¼-inch black urethane mat lightly sanded with medium grit sandpaper to increase friction.

The fluorescent dye used for the mouse and hand datasets was Opticz Bright Red UV Blacklight Reactive Invisible Ink (DirectGlow, DGINK1OZR).

### Mouse diverse dataset collection
Data were obtained from adult C57BL/6 male and female mice (8–12 weeks old) housed on a 12:12 h light cycle at 65–75 °F with 40–60% humidity. Training data and uniform and diverse held out data were collected from 2 male and 2 female mice. Challenge data consisted of videos of 14 male and female mice (exact sex and age distribution is unknown as these videos were randomly selected from archival videos across labs).

Prior to video capture, animals were anesthetized with 1–3% isoflurane and a felt-tipped marker was used to apply fluorescent dye to the target region. In all experiments reported, the entire hand or foot, both ventral and dorsal, was coated in dye up to the wrist or ankle. The dye was allowed to dry for 5 min prior to video recording, and the animal was visually checked under UV illumination for stray dye outside of the target region.

Mice were recorded performing three behaviors over two days: reaching, string pull, and free movement. For the reaching task[8], the training protocol consisted of placing the mouse in a 20 cm tall × 8.5 cm wide × 19.5 cm long clear acrylic box with an opening in the end wall measuring 0.9 cm wide and 9 cm tall. A 3D-printed, 1.8 cm tall pedestal designed to hold a food pellet (20 mg, 3 mm diameter; Bio-Serv) was placed 1 cm away from the front of the box opening and displaced 0.5 cm lateral to the opening and opposite the preferred reaching forelimb. Food pellets were placed on top of the pedestal as the reaching target. Mice were food restricted to ~85% of their original bodyweight and trained to reach for food pellets for either 20 min or until 20 successful reaches (defined as pellet retrieval) were accomplished. For the string pull behavior[65], animals were food restricted as described above. Ten to twenty strings were suspended above an open cage; one end of the string was placed within reach of the animal, and the other end was attached to a food reward. Animals quickly learned to pull the string into the cage to receive the reward. For the freely moving behavior, the animals did not receive any training or food rewards and were allowed to roam freely within the enclosure.

For all behaviors, cameras were triggered at a frame rate of 200 Hz and a resolution of 848 × 848 under alternating visible and UV illumination, and temporally adjacent frames were grouped into pairs. In both the biphasic and triphasic illumination schemes, the cameras and lights were triggered on a 10 ms cycle. For biphasic, UV illumination was on from 0 to 5 ms while visible illumination was on from 5 to 10 ms. Camera shutters were triggered at 2.5 ms and 7.5 ms, and the exposure duration was 2 ms. For triphasic, all time parameters were identical to biphasic with the exception that UV illumination was extinguished at 2.5 ms. For the freely moving behavior, to decrease data bandwidth,

only 1 out of every 50 image pairs was recorded to disk (2 pairs per second). For the reaching and string pull behaviors, all image pairs were recorded at the full rate of 100 pairs per second, but to decrease data bandwidth, recording was only activated by the experimenter during individual reaches or string-pulling bouts. To decrease the number of frames missed due to the delay between behavior onset and the experimenter starting the video recording, image pairs were continuously saved to a 100-pair buffer in memory that was written to disk upon activation of video recording.

The uniform test set contained 1231 frames. The diverse test set contained 4438 test images, half of which were manually annotated with bounding boxes to compute the OKS-mAP metric (Supp. Table 1), and 600 of which were manually labeled with keypoint locations to compare fluorescence-derived labels with manual ground truth (Supp. Fig. 3a). The OKS-mAP metric was computed using the COCO Analyze python package[31].

## Mouse challenge dataset collection

To create the challenge dataset, we sampled data from a variety of archival datasets. Represented in these datasets were videos of male and female mice from two different labs (Azim and Goulding), five different behaviors (pellet reach, head-fixed water reach, treadmill, balance beam, string pull), both color and grayscale imagery, and an approximate image scale range of 25-160 pixels per centimeter. All data had been collected under different lighting conditions and from different cameras than the diverse dataset (total of ten visual environments corresponding to different configurations of cameras and lights). The water reach, treadmill, string pull, and balance beam behaviors had been captured on behavioral apparatuses different from those represented in the diverse dataset.

Each archival dataset contained a different number of videos of different lengths. Simply combining all datasets would have resulted in a highly skewed number of frames per dataset. Therefore, we sampled 4-16 clips per dataset, depending on the specific video sizes, to achieve a more balanced number of frames from each dataset. From among those clips, we then sampled 660 random images to receive ground truth labels.

To produce ground truth labels for the challenge set, we manually labeled the sampled images using LabelStudio[66], an open-source web application. Each frame was labeled by two annotators. Annotators were instructed not to guess the location of any landmark unless it was obvious. If a landmark was out of the camera's field of view or occluded such that it was impossible to know its precise location, then the annotator was instructed to record that landmark as absent. Only frames for which both annotators marked the landmark as absent or the annotators' labels were within 5 pixels of each other were retained in the final label set. The two annotators' landmark locations for each frame were averaged to produce the final ground truth location. After the label reconciliation process, there were 612 labeled images in the challenge set.

## Deep learning model training

The deep learning model used for all experiments (both mouse and human hand) was the DeeperCut model, which is based on the ResNet-101 architecture pre-trained on ImageNet and fine-tuned on the given training dataset[10,30,67]. We trained each model for 4.12 million training iterations with an image pre-scaling factor of 0.8. These parameter values were each selected using 1-D grid search. All other training parameters were set to their default values[30]. The stride of the neural network was 8, yielding output heatmaps and refinement maps 8-10 times smaller in each dimension than the input images (for example, the network produced 86 × 86 square output maps for the 848 × 848 square training images). During training we performed more extensive data augmentation than the original DeeperCut implementation. Prior to each training image being consumed by

DeeperCut, it was randomly perturbed using the "imgaug" Python package. To each training image we randomly applied Gaussian blur (sigma between 0.0 and 0.5), random contrast scaling (scale factor between 0.75 and 1.5), additive Gaussian noise (sigma of 5% maximum intensity), random channel scaling (scale factor between 0.8 and 1.2), random cropping (0–10%), random affine warping (scale 0.8–1.2, translation 0.8–1.2, rotation +/− 15 degrees, sheer +/− 8 degrees), and conversion of color images to grayscale with 50% probability.

For the interleaved training procedure (Supp. Fig. 4), the multi-landmark neural network was trained on a dataset consisting of 126,920 images in which the mouse foot was fluorescently labeled, along with the 388,496 images from the diverse dataset in which the mouse hand was fluorescently labeled. The DeeperCut objective function was modified such that, for each image, only the confidence output map for the landmark labeled in that image contributed to the objective function. For instance, for an image in which the foot was labeled, the confidence output map for the hand did not contribute to the objective. All other training parameters remained the same as for the other models trained on the diverse training sets.

For the human hand dataset, we computed a fluorescence mask to segment the hand from the background and augmented the training data with random synthetic background imagery. The random background was generated via the composition of several noise functions producing both high and low spatial frequency noise.

## Label post-processing and real-time optimization

Scale optimization was performed at test time using 1-D hierarchical grid search with two levels. First, the objective was computed over a coarse range of scales ($2^k$ for k ranging from -1 to 1 in increments of 0.5), and then over a finer range of scales centered on the optimal value from the previous level ($2^{k+j}$ for j ranging from −0.33 to 0.33 in increments of 0.16). We experimented with two proxy metrics: smoothness, as measured by the average landmark displacement distance between adjacent frames (i.e., average speed); and confidence, as defined by the average confidence of the model's predictions over the entire clip. We found that occasionally the smoothness metric failed to capture the accuracy of the trajectory; sometimes highly smooth trajectories were nevertheless spurious. The confidence metric did not appear to suffer from this type of failure. Therefore, all experiments presented were performed using mean confidence as the objective function being optimized. For image-level scale optimization, this search procedure was repeated independently for each image. For clip-level scale optimization, the procedure was performed once for all images in a video.

To enable real-time visualization for interactive adjustment of image capture, we developed a user interface in which the live video is displayed to the user with the model's predicted landmark label overlaid, alongside a plot of the landmark's most recent 100 x positions, y positions, and confidence values over time. The plot is continuously updated in real-time with an average throughput of approximately 10 frames per second, allowing the user to adjust the position of the animal, camera, and lighting interactively.

## Visual barcoding

Dye was applied to the human hand using a brush to aerosolize it into a fine mist: a stiff toothbrush was loaded with dye, the bristles were agitated, and the resulting mist was allowed to settle on the target object. This procedure was conducted under UV illumination to monitor coverage of the target, and the dye was allowed to dry for five minutes. There was only one human (male) participant, so disaggregation by sex and/or gender was not relevant. Consent to publish information potentially identifying the individual was obtained.

Video data were collected using the same video capture setup described above from one hand. OpenCV[37] was used to compute SIFT features and perform SIFT matching. SIFT features were computed

with a contrast threshold of 0.01, and all other parameters were set to their default values.

To reduce the number of spurious matches, we performed spatial coherency filtering. In our filtering algorithm, for a putative match (*a*, *b*) to be considered valid, it must have at least one coherent neighboring match. A neighboring match (*c*, *d*) is considered coherent if the left endpoints *a* and *c* are within a distance of δ and *b* and *d* are within a distance of ε. We set δ and ε equal to 50 pixels for our experiments. Neighboring matches were computed efficiently using the KD tree implantation from the "scipy.spatial" Python package[68].

When selecting template atlas images, an algorithm was used to rank all training images. Our ranking algorithm is iterative and greedy; it repeatedly selects the next best image from the training set. Specifically, it selects the training image that matches the greatest number of unmatched SIFT features in the training set. Then, all SIFT features that the selected image matches are marked as having been matched. We implemented the algorithm by modifying published C++ code[69] to perform large-scale approximate SIFT matching using a vocabulary tree[70]. Specifically, we changed the vocabulary tree implementation so as to compute the simple weighting scheme described above rather than the more complex TFIDF scheme from the original paper[70].

First, SIFT features are computed for all images in the training set and grouped into 500,000 clusters using an approximate k-means algorithm. Each feature is assigned a weight equal to the number of features in its cluster. Then, each image is assigned a weight equal to the sum of the weights of its features; the image weight represents an estimate of the number of features across the entire dataset that would be matched by that image. The image with the greatest weight is the next image selected by the algorithm. Finally, the weights of features matched by the selected image are decremented, all image weights are updated, and the next image is selected. We automatically selected the top 100 template candidates using this procedure, and manually refined the list down to 10 final template images covering the palm of the hand. We then used Adobe Photoshop to create 12 binary masks for each template image, indicating the extent of the 12 finger segments in that template.

To visualize and quantify the number of SIFT matches, we started with a video clip of length 1000 frames containing no template images. We computed the number of matches between each template image and each frame of the clip. Then for each finger segment, we computed the total number of matches across all templates. This procedure resulted in 12 numbers representing match counts for each finger segment in each frame of the clip. Finally for each frame, we overlaid a scatter plot with the matches from each template segment in a different color.

The manual neighborhood selection pipeline for generating labels is very similar to the process described above. The only difference is an additional step in which all matches corresponding to a segment are pooled together to produce a label for that segment. Pooling was performed in the following way: if the segment had at least 5 matches, then the median of the x-values and y-values were used as the x and y position of the label, respectively. If the segment had fewer than 5 matches, it was recorded as absent. Labels produced in this way were then used to train a DeeperCut model to predict the location of the 12 finger segments. For evaluation, ground truth labels were generated manually with LabelStudio.

The automatic neighborhood selection pipeline is structured as follows. First, a single template image was selected as the base image. A subset of SIFT features in the base image were selected by iteratively selecting one SIFT feature and then removing all other features within 30 pixels (using a KD tree). Seeds were selected randomly and the minimum-distance requirement ensured they were spaced apart. The points within a certain radius of each seed were then treated as a cluster. Therefore, some points belong to multiple clusters. This approach resulted in a set of template features with a minimum spacing of 30 pixels that form the set of landmarks tracked by the deep

learning model. For each video frame in the training set, SIFT matches were computed between the base image and the target frame. For template features that matched the target image, the position of the match in the target image was used as the landmark position. If a particular template feature failed to match the target, then neighboring features within a radius of 30 pixels that had successfully matched were used as proxies; those feature matches were used to fit a local homography using OpenCV[37]. That homography was then used to transform the position of the original template feature from the base image into the target image.

### Reporting summary

Further information on research design is available in the Nature Portfolio Reporting Summary linked to this article.

## Data availability

All data supporting the findings described in this manuscript are available in the article and the Supplementary Information. The main image datasets that support the findings of this study are publicly available online (https://cnl.salk.edu/~dbutler/glowtrack). Source data are provided with this paper.

## Code availability

Materials available on a public Github repository (https://github.com/azimlabsalk/glowtrack) include: design files for dome fabrication; image capture code; Arduino control code; software packages for processing strobed fluorescence video, generating motion capture models, and evaluating model performance; and a GUI for real-time viewing of strobed fluorescence video and motion capture model output. This code repository has been archived[71]. Correspondence and requests should be addressed to E.A. (eazim@salk.edu).

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

## Acknowledgements

We are grateful to E. Sanders, A. Thanawalla, and N. Baltar (Salk Institute) for providing trained mice and archival video datasets; G. Gatto (Salk Institute, Uniklinik Köln), Martyn Goulding (Salk Institute), and K.W. Huang (Salk Institute) for providing archival video data; P. Nguyen (Salk Institute) for assistance with mouse husbandry and lab operations; A. Aeruva, N. Benhaim, H. Gao, J. Khatibi, M. Ochoa, P. Nguyen, D. Saklaway, and G. Salmun (Salk Institute) for labeling images; N. Baltar (Salk Institute) for assistance with building the dome; K. Cortes (Salk Institute) for data collection; A. Cao (Salk Institute) for the mouse and human hand illustrations; and B. Brunton, P. Karashchuk, U. Manor, T. Pereira, J. Tuthill, and members of the Azim lab for valuable discussion and comments on the manuscript. The work was supported by the UCSD CMG Training Program and a Jesse and Caryl Philips Foundation Award (A.P.K.); and by the National Institutes of Health (R00NS088193, DP2NS105555, R01NS111479, RF1NS128898, and U19NS112959), a Salk Institute Innovation Grant, the Searle Scholars Program, The Pew Charitable Trusts, and the McKnight Foundation (E.A.).

## Author contributions

Conceptualization and design: D.J.B., E.A.; Data collection: D.J.B., A.P.K.; Hardware development: D.J.B., A.P.K.; Software development: D.J.B.; Algorithm evaluation: D.J.B., S.R.; Data analysis: D.J.B.; Manuscript preparation: D.J.B., E.A.

## Competing interests

The following authors, D.J.B., A.P.K., and E.A., declare the following competing interests: a patent application filed by The Salk Institute: US-20220121878-A1. The following author, S.R., declares no competing interests.
