## [Peer Review File · Nature Communications]

REVIEWER COMMENTS

Reviewer #1 (Remarks to the Author):

Butler et al. present a novel idea to the pressing need for generating labels to perform deep learning of marker less motion capture.

The approach presented by the authors is clever and highly innovative as it allows for the non invasive generation of datasets that go far beyond what most current datasets offer. Further, the approach offers an opportunity into the direction of true generalizable models instead of the one off approach that is currently used.

The reviewer is especially fascinated by the simple trick of using triphasic illumination to counteract natural reflectance within the setup, something that can be extremely detrimental to even marker based systems used within enclosures that contain reflective PVC for example.

The manuscript contains many nice thoughts on improving and importantly robustly and comprehensively evaluating the performance of a deep learning model on animal tracking. For example, the idea of rescaling, especially optimized rescaling is an important characterization of the performance of the models we use and is something that is often overlooked. In this context it would be crucial for the authors to acknowledge the vast computer vision literature on human key point estimation that has worked on these problems before and proposed many other often overlooked algorithmic approaches, especially related to image augmentation. In essence the problem boils down to this: How can we learn a shared representation across modalities, viewpoints and conditions.

For the serial approach of the manuscript the authors avoid (in my humble opinion) to discuss the elephant in the room: Can we potentially use this simpler approach to label one key point after another on the same animal and scale up a full skeleton that way in a round robin kind of way. Lets say one trains a model for hand a model for foot a model for head and so forth and then generate a combined large label datasets to train a combined model. How much better would the performance really be when comparing it to approaches of large datasets from Dunn et al. or Bala et al.?

As an alternative strategy, the authors present their parallel speckle approach that allows for SIFT feature matching and multiple key points to be labeled and learned simultaneously. As an example they use a human hand and the results are certainly convincing. Here is the problem however, it is very unclear from the current manuscript if this approach easily generalizes to entire bodies of animals or if it is inherently limited to smaller, more detailed features. Both are fine in and on themselves but need to be addressed and clearly spelled out in the manuscript. It is also a bit inconsistent that one organism is

human the other animal. While that shows some set of diversity a combination of both approaches to a more comprehensive dataset for either organism would have been preferable from this reviewers perspective.

Overall I think this is a very nice manuscript that falls short of its ultimate promise a little but demonstrates a clear trend in the right direction. The authors should address these shortcomings more clearly which would make me very supportive of the manuscript.

Minor comments:

1. Figure 2 would be more illustrative if the example shown in panel a corresponded to the raw images presented in the other panels.

Reviewer #2 (Remarks to the Author):

In the submitted work, the authors present a method for automatically creating large amounts of training data for machine learning-based pose estimation methods. Large-scale data is especially useful for training robust and generalizable machine learning models. In order to reduce the manual labor of data annotation, the authors propose the usage of UV fluorescent dye that is applied to the target and only visible under UV light. In this work it is shown for a rodent hand and human digits. The authors use a custom multi-camera setup with UV and visible light illumination that are alternatively switched on, thus making the dye only visible in the UV image. For extracting the labels, they propose two methods: a serial and a parallel labeling approach, suitable for extracting single landmarks and dense labels, respectively. The results show that it is feasible to automatically create accurate training data using the proposed method, and to train an existing machine learning method (DeeperCut [30]) for pose estimation that is able to accurately localize the landmarks and is robust to different laboratory setups.

Overall, the paper is easy to follow and well structured. The language is appropriate and easy to understand. The authors added several images and figures that help the reader understand the paper. It is highly appreciated that the authors make their code and data publicly available.

In terms of novelty, the presented method pushes forward the possibilities of conducting and simplifying animal studies. It can have large practical impact, and the scientific novelty seems sufficient.

The use of UV dye and using a triphasic illumination together with the hardware setup is innovative. Some weaknesses are in the parallel labeling approach and the choice of the machine learning method that need to be justified. Overall, more effort should be spent on the practical considerations, and several remarks should be addressed in a revision of the papers, as detailed below.

Major remarks:

- Why did the authors choose the DeeperCut [30] model as pose estimation method? For the problem at hand (single subject pose estimation), this model seems like an overly complex method, as it was designed for multi-person pose estimation and to explicitly handling the pairwise relationship between joints, which is not even possible for the serial labeling use-case. Overall, a much simpler neural network model would be sufficient, as e.g. [Sun et al.: Deep High-Resolution Representation Learning for Human Pose Estimation, Conference on Computer Vision and Pattern Recognition, 2019] or [Newell et al.: Stacked Hourglass Networks for Human Pose Estimation, European Conference on Computer Vision, 2016]
- More details regarding the neural network could be helpful: What is the runtime for single frame inference? Is it possible to run the pose estimation in real-time (esp. regarding the feedback experiment for tuning the parameters)? What is the inference resolution of the neural network, ie. the spatial localization accuracy of the landmarks, which is usually lower than the processed camera image? What are the training details of the neural network (deep learning framework, learning rate, batch size, online augmentation, etc.), or is everything kept the same as in the DeeperCut [30] code?
- Sec. 2: Are there size limitation for applying the dye? Is it possible to label each digit of the rodent hand?
- Sec. 2: The proposed camera dome allows capturing different viewpoints simultaneously. Does this generalize to arbitrary camera positions, or do the positions still need to be close to the camera poses of the dome?
- Sec. 2.2: How does the network generalize to different mice, i.e. gender, age, race, fur color, etc.? This is essentially related to a leave-one-subject-out validation.
- Sec. 2.3: The authors mention to "label one hand on the mouse" in the serial labeling use-case. It could be more practical to derive labels for all limbs or at least both hands. Is this possible with that approach?
- L107: The authors denote that they are able to label "nearly any landmark", which sounds verbose. What are the actual limitations?
- Fig. 4: The figure shows the trajectory of the landmark and the ground truth in order to visualize the accuracy. How does this capture the 2D landmark location in the image with this 1D plot? It would be more appropriate to plot the Euclidean error between prediction and ground truth as curve, in order to show the accuracy and not both curves superimposed.
- Sec. 2.6: Why was SIFT chosen, as it is known to lack repeatability on almost texture-less surfaces such as the palm/digits? How does the visual bar-code react to non-rigid deformation? The used homography assumes local planarity. Would it make more sense to use an optical flow-based method, e.g. [Liu et al.:

SIFT Flow: Dense Correspondence across Scenes and Its Applications, Transactions On Pattern Analysis and Machine Intelligence, 2011], for tracking the points compared to the proposed homography?

- Sec. 2.7: How are the 50 different landmarks represented in the machine learning method (DeeperCut)? What is the benefit over using the centroid only? How does the over-parameterized pose with 50 landmarks compare to per-pixel classification, ie. semantic segmentation?

- Sec. 2.7: The authors describe the manual method for parallel labeling in great length, however, the fully automatic method is more relevant to the actual use-case and it could be sufficient to describe that solely. Does it make sense to save space and significantly shorten the manual method for parallel labeling?

- Sec. 2.7: From the text, it is unclear how the regions/centroid/clusters are initialized for the automatic method. Are the initial locations random and then further clustered?

- L477: The authors propose a semi-automatic method for the dense labeling problem, "because a human annotator cannot be given a simple description of each landmark". However, such a manual labeling was done e.g. in [Guler et al.: Densepose: Dense human pose estimation in the wild, Conference on Computer Vision and Pattern Recognition, 2018].

- L654: The authors marked the landmark as present only if "the annotators' labels were within 5 pixels of each other". This could create false negatives, and as the authors denote before that images without labels help boost the model performance. Did the authors notice any drawbacks by using this strategy?

- Sec 4.4: The authors mention an "image pre-scaling factor of 0.8". How does that change the image, or is this a simple resize operation? Further, the authors mention a "conversion of color images to gray-scale with 50% probability". Why is that done, and why do the authors not provide a separate model for color and gray-scale? How is the 1-channel gray-scale (intensity) image converted to a 3-channel color (RGB) image? Why is so much attention spent on red illumination (e.g. Supp. fig. 1)?

- References: Please use a consistent format. Esp. [13,38,41] consider replacing bioRxiv with proper conference/journal publication, [14,15,40] pages are missing, [19] pages contain strange string, [26,28,31] for conference use consistent usage of pages or not print pages at all, [28,31,39,43,61] avoid an additional conference abbreviation, [30,60] citations are incomplete, etc.

- Supp. fig. 6: The authors mention the use of "morphological erosion". However, the images look non-binary, but the morphological operations for foreground/background segmentation are only well-defined on binary images, esp. when blending in different backgrounds. Please clarify.

Minor remarks:

- L164: Consider using consistent naming: "signal to noise" -> "signal-to-noise ratio" as in L149

- L171: Instead of "islands" consider using "blobs" as the more relevant image processing term.

- L181ff: Capitalization "section" -> "Section"

- L208: Typo "unform" -> "uniform"

- L456: The punctuation is confusing when listing the errors of the different training. Please check usage of "," and ";"
- L510/513: Consider replacing "many-camera setup" with "multi-camera setup" to be more consistent with the remaining text.
- L659: There are Sections 4.4/4.6, but no 4.5!?
- L936: "Video (10x speed)" suggests a fast-forward video, but actually it is slowed down.
- Supp. video 1: The labels of the camera feed "UV" and "visible" seem swapped.
- Supp. video 3: What do the numbers next to the detected landmark denote?
- Supp. video 5: Without reading the paper beforehand, the dots are not self-explanatory and a short explanation could help the reader.

Reviewer #3 (Remarks to the Author):

The authors describe two approaches to automatically label keypoints on an animal in video frames to serve as ground truth for pose estimation algorithms. In the first approach, they paint a single landmark (a mouse hand) using UV fluorescent probe, and use an 8 camera setup, multiple UV light sources, and a triphasic data collection regime to record the position of the painted region with high signal to noise. They demonstrate that this approach can be used to collect a reasonably large training dataset (~300 k frames), across three different behaviors, and that the hand can be tracked using a model trained on the diverse training dataset, but not one from a single camera view. They then show that the diverse training dataset produces a moderate-quality hand tracker on out-of-domain images, after applying some additional algorithmic refinement by resizing the input images by scale factors optimizing model confidence. In the second approach, they apply the fluorescent label to the human hand, essentially as an added source of texture, and present several strategies for training keypoint detectors to locate regions of interest in these blobs.

Producing more generalizable pose tracking models is important for the life sciences, and automated ways of collecting training data are a key step in that direction. I think the approach of UV labels is clever, and I like the half-dome optical setup and approach for illumination randomization. I think the result showing the enhancement of tracking with added labels is useful, and there are multiple technical details, e.g. clip-scale optimization that others in the field may find useful. While I'm not sure that fluorescent labeling will become widespread in the field, I do think this manuscript contains multiple innovations that people in the field will find useful. That said, I do think the manuscript suffers from several weaknesses. First, I think several of the claims are overstated, mostly in regards to generality of tracking from models. Second, while the manuscript claims the serial tracking approach can be used to track multiple points, evidence is not provided. Third, the parallel tracking approach is comparatively under-contextualized and benchmarked, and I am not convinced of the results.

Overall, I am not against publication of this manuscript, but it needs revision.

Major

1) What is claimed:

L497: “Together, the generalizability and scalability of these approaches enable automated pose detection and kinematic quantification across species with less effort, better accuracy, and higher resolution than is feasible with manual annotation approaches.”

“Together, these results demonstrate that [...], and that our fluorescence labeling approach can provide the necessary quality and quantity of training data.”

I don't believe these are supported by the data. I do think that the advances in the paper are useful and meaningful to the field. But the performance of the generalized rodent hand tracking in figure 4e falls short of the manual annotation approach in 3g (I realize these are different test domains). I do think that these approaches can facilitate generalization and perhaps less need for fine tuning, but I don't think it replaces them. Same comment for accuracy and resolution. Similarly, the ability of this approach to perform pose tracking has not been established (only a single point shown). There are multiple comments about kinematic quantification in the manuscript, but the only example is figure 4b, which is more a demonstration of error reduction.

2) Serial tracking approach. The authors haven't demonstrated serial application. I think this is likely feasible, but it needs to be shown. It is possible that there will be eg swaps across the hands and feet.

3) Contextualization, exposition, and benchmarking of the parallel tracking approach. This work was comparatively underdeveloped to the serial tracking approach. There is a schematic algorithm, and some quantification, but the 1) contextualization within computer vision 2) exposition and 3) quantification is lacking.

- What is the relationship between this approach and other approaches in computer vision for automated keypoint discovery. You can imagine taking the same approach as used here, but using textural features or high contrast features directly from CV. Work in stereopsis has pursued this (e.g. KeypointNet) and likely monocular CV. Moreover, in the domain of hand tracking there are approaches that use motion capture as automated labels (e.g. Shangchen Han, ..., Robert Wang, ACM transactions Graphics 2020). Moreover, some of the work with automated neighborhood selection touches on surface tracking and it would be good to at least acknowledge this connection.

- Can you include a workflow of the algorithm.
- How does this SIFT approach compare with using features from computer vision within the same pipeline. Is it a strength that this does not e.g. generalize to tracking on the top of the hand?
- There needs to be some sort of yardstick baseline/sanity check for these approaches that can give a sense of the significance of the precision-recall curves in 6e,6k. These curves give me very little sense of the true performance, as they depend heavily on the types of images in the test set, which could be very in-domain and a simple e.g. regression approach based on CV features might do just as well. This approach certainly doesn't have broad generalization across hand pose, image scale, etc (the reason for many
- L485 'Using this approach, our model could reliably identify dozens of landmarks on a hand'. don't believe the claim of 50 simultaneously tracked keypoints is every quantified. Given the keypoint error is ~40 px (Figure 6k) I am not sure there is enough space on the hand/finger to simultaneously detect such a large number of keypoints. Given the large amount of jitter in the supplemental video, I don't believe it is accurate to say these are uniquely identified.
- Nits: scale bars on the images are needed (how big are 40 pixels?). I couldn't find a description of the deep learning model used for hand tracking here, hyperparameters, etc.

4) Generalization in the challenge dataset: How well does the approach generalize to other behaviors (eg rotarod?). Many of these domains are quite similar behaviorally.

Minor

- Illumination randomisation vs. other image augmentation during training
- L29 "generating hidden labels free of human error using fluorescent markers'. Humans choose the position of labels, so these are very dependent on human choices.
- What body structures can this approach be applied to? Is it only skin? There are some comments about fur in the conclusion but I do not believe the limitations of the existing approach were clearly stated.
- The manuscript but feels very long. Some of the technical details could be moved to supplement. The parallel tracking section especially.
- The comparison of manual vs fluorescence labels in supp figure 3a is surprising. I would imagine that there are a subset of frames that are visible in RGB, but lack SNR in dye to get detected (eg due to self-

occlusion of the UV source). What fraction of frames is this and is the test set here randomly sampled or restricted to frames in which there is UV ground truth?

- Section 2.3, l237. The term 'manual' here is confusing, since these are not actually manually labeled.
- Figure s3b – were the number of frames here balanced across conditions (e.g. of lighting/viewpoint).
- Diverse dataset pretraining. How does pretraining on other action datasets, e.g. COCO compare to the rodent dataset?
- Figure 4: Will clip level optimization increase the false positive rate of the approach? Do all images in the test set contain hands?
- Why stop at 300 k frames in the diverse training set?
- L399 'optimal' number of templates – is there justification for why 10, and not 20, 30, 40, etc. is optimal?
- L436 – a diagram of the network architecture would help clarify the training procedure

We are enclosing a revised version of our manuscript “Large-scale capture of hidden fluorescent labels for training generalizable markerless motion capture models”.

We have modified the manuscript extensively in response to the constructive comments. Each issue raised has been addressed directly, through the inclusion of new experiments, data analysis, and/or through clarification of text and concept. Major additions include: a demonstration that the one-landmark-at-a-time serial labeling approach can be used to train a multi-landmark model (using a new interleaved training approach), an evaluation of an additional neural network architecture, more detailed description of image set diversity, and modified text and additional figures to improve the presentation and interpretation of our findings.

We include below a detailed account of the specific responses to each of the Reviewers’ comments in the order they were made, and all changes in the original manuscript are tracked in **red**.

Reviewer 1

Butler et al. present a novel idea to the pressing need for generating labels to perform deep learning of markerless motion capture.

The approach presented by the authors is clever and highly innovative as it allows for the noninvasive generation of datasets that go far beyond what most current datasets offer. Further, the approach offers an opportunity into the direction of true generalizable models instead of the one-off approach that is currently used.

The reviewer is especially fascinated by the simple trick of using triphasic illumination to counteract natural reflectance within the setup, something that can be extremely detrimental to even marker-based systems used within enclosures that contain reflective PVC for example.

... Overall I think this is a very nice manuscript that falls short of its ultimate promise a little but demonstrates a clear trend in the right direction. The authors should address these shortcomings more clearly which would make me very supportive of the manuscript.

Comment 1. *The manuscript contains many nice thoughts on improving and importantly robustly and comprehensively evaluating the performance of a deep learning model on animal tracking. For example, the idea of rescaling, especially optimized rescaling is an important characterization of the performance of the models we use and is something that is often overlooked. In this context it would be crucial for the authors to acknowledge the vast computer vision literature on human key point estimation that has worked on these problems before and proposed many other often overlooked algorithmic approaches, especially related to image augmentation. In essence the problem boils down to this: How can we learn a shared representation across modalities, viewpoints and conditions.*

We appreciate the Reviewer's comments and agree that it would be helpful to include more background information about prior work on key point estimation, especially with regards to image rescaling and augmentation. One line of particularly relevant research involves a procedure called test-time augmentation (Krizhevsky, Sutskever et al. 2017, Kim, Kim et al. 2020, Shanmugam, Blalock et al. 2021) (a reference list is included at the end of the letter). Like our method, test-time augmentation involves applying a neural network to different transformed versions of a test image and combining the resulting predictions to yield a single higher-accuracy prediction on the original image. Much work in test-time augmentation has focused on the single-image case, where, unlike with our data, information is not shared across the frames of a video. To combine the multiple predictions produced for each image, various strategies have used either the mean (Krizhevsky, Sutskever et al. 2017, Kim, Kim et al. 2020) or a learned function (Shanmugam, Blalock et al. 2021). We have added this additional information and associated references to the manuscript (**pages 3031**).

Comment 2. *For the serial approach of the manuscript the authors avoid (in my humble opinion) to discuss the elephant in the room: Can we potentially use this simpler approach to label one key point after another on the same animal and scale up a full skeleton that way in a round robin kind of way. Let's say one trains a model for hand a model for foot a model for head and so forth and then generate a combined large label datasets to train a combined model. How much better would the performance really be when comparing it to approaches of large datasets from Dunn et al. or Bala et al.?*

We thank the Reviewer for the comments and agree that training a network on multiple landmarks in a round robin fashion using serial labeling would provide a straightforward approach to scale up keypoints across the body. One obvious, though perhaps clunky, way to do this would be to just train a different network to detect each keypoint of interest and then run detection using each trained network separately. While this approach works, it would become quite onerous as the number of landmarks increases.

A more elegant approach would be to train a single model to detect multiple landmarks using our serial labeling approach. To demonstrate feasibility, we have performed **new experiments** in which we trained a single network to detect hand and foot landmarks using serial labeling. The modified training procedure is explained in the first two panels of a **new Supp. Fig. 4a,b** and described on **pages 19 and 37**. Briefly, we implemented a method, which we call interleaved training, in which only the landmarks that are labeled in a particular training image contribute to the loss function (**Supp. Fig. 4b**). We used fluorescence labeling to collect an additional training dataset (with 127,000 images) in which the foot of the mouse was labeled. A combined, multi-landmark model trained on both the mouse hand and foot performed similarly to two separate models trained on the hand and foot respectively (**Supp. Fig. 4c**), showing that the interleaved training scheme can be used to build up multi-landmark models from single-landmark datasets.

A direct head-to-head comparison between our approach and other recent approaches, like those by Dunn et al. or Bala et al., would be desirable, but there are technical details that make such a direct comparison not feasible. In particular, the comparison is greatly complicated by the different species involved (in the case of Bala et al.) and the different landmarks and behaviors under study (in the case of Dunn et al.). Thus, the complexity of reproducing the exact context of the other papers (species, landmarks) or, alternatively, comparing raw numerical error metrics between the methods despite dramatic differences in experimental conditions, would make the results difficult to interpret.

Comment 3. *As an alternative strategy, the authors present their parallel speckle approach that allows for SIFT feature matching and multiple key points to be labeled and learned simultaneously. As an example*

they use a human hand and the results are certainly convincing. Here is the problem however, it is very unclear from the current manuscript if this approach easily generalizes to entire bodies of animals or if it is inherently limited to smaller, more detailed features. Both are fine in and on themselves but need to be addressed and clearly spelled out in the manuscript. It is also a bit inconsistent that one organism is human the other animal. While that shows some set of diversity a combination of both approaches to a more comprehensive dataset for either organism would have been preferable from this reviewer's perspective.

The Reviewer raises an important point that requires clarification. Applying the parallel labeling approach to the mouse is certainly a direction we plan to take this method. Our preliminary experiments in that direction revealed that problems with movement of the fur, meaning barcodes do not remain stationary, is a challenge we need to overcome. With this manuscript, we present proof-of-concept for both the serial and parallel labeling pipelines, noting important overlap between data collection approaches. We feel that the human hand dataset demonstrates the potential of the parallel labeling approach on highly articulated objects with relatively smooth surfaces, and more generally, the use of both mouse and human in the paper emphasizes the species generalizability of our approaches. The serial pipeline could certainly be applied to the human (or any other animal where a landmark can be labeled with dye). Applying the parallel labeling pipeline to an animal with fur is an important goal and, while outside the scope of our proof-of-concept demonstration in this paper, it is one we plan to tackle moving forward.

Minor points

Comment 4. *Figure 2 would be more illustrative if the example shown in panel a corresponded to the raw images presented in the other panels.*

We should clarify: **Fig. 2a** is a raw image – the red color comes from the dye rather than from image processing. The reason that the images in panel **a** are in color while the images in panels **b & c** are monochrome is to highlight different things: the problem of background noise in fluorescence images is most acute in monochrome images, whereas the content of our visible-light images is most obvious in color. We now clarify this point in the figure legend.

Reviewer 2

In the submitted work, the authors present a method for automatically creating large amounts of training data for machine learning-based pose estimation methods. Large-scale data is especially useful for training robust and generalizable machine learning models. In order to reduce the manual labor of data annotation, the authors propose the usage of UV fluorescent dye that is applied to the target and only visible under UV light. In this work it is shown for a rodent hand and human digits. The authors use a custom multi-camera setup with UV and visible light illumination that are alternatively switched on, thus making the dye only visible in the UV image. For extracting the labels, they propose two methods: a serial and a parallel labeling approach, suitable for extracting single landmarks and dense labels, respectively. The results show that it is feasible to automatically create accurate training data using the proposed method, and to train an existing machine learning method (DeeperCut [30]) for pose estimation that is able to accurately localize the landmarks and is robust to different laboratory setups.

Overall, the paper is easy to follow and well structured. The language is appropriate and easy to understand. The authors added several images and figures that help the reader understand the paper. It is highly appreciated that the authors make their code and data publicly available. In terms of novelty, the presented method pushes forward the possibilities of conducting and simplifying animal studies. It can have large practical impact, and the scientific novelty seems sufficient. The use of UV dye and using a triphasic illumination together with the hardware setup is innovative. Some weaknesses are in the parallel labeling approach and the choice of the machine learning method that need to be justified. Overall, more effort should be spent on the practical considerations, and several remarks should be addressed in a revision of the papers, as detailed below.

Comment 1. Why did the authors choose the DeeperCut [30] model as pose estimation method? For the problem at hand (single subject pose estimation), this model seems like an overly complex method, as it was designed for multi-person pose estimation and to explicitly handle the pairwise relationship between joints, which is not even possible for the serial labeling use-case. Overall, a much simpler neural network model would be sufficient, as e.g. [Sun et al.. Deep High-Resolution Representation Learning for Human Pose Estimation, Conference on Computer Vision and Pattern Recognition, 2019] or [Newell et al.. Stacked Hourglass Networks for Human Pose Estimation, European Conference on Computer Vision, 2016]

We thank the Reviewer for the comment, and we agree that DeeperCut is a somewhat complex model and that simpler models are likely sufficient. In fact, to decrease complexity we only used one component of the DeeperCut model (the heatmaps and refinement maps), not the whole algorithm in its full complexity. We now clarify this point on **page 10**.

We chose DeeperCut for a few reasons. First, DeeperCut is the architecture used by DeepLabCut, arguably the current standard in animal behavioral tracking, offering the most direct way for potential end-users around the world to apply our methods. Second, we considered using the Stacked Hourglass architecture (Newell et al.), but the implementations we found were not fully convolutional and therefore couldn't process different sizes of input image. Therefore, our experiments testing generalizability among different experimental setups and labs were more straightforward to conduct with the DeeperCut network, which is fully convolutional and can handle arbitrarily sized input images.

That said, to address the Reviewer's concerns, we have now also trained a neural network with a different architecture. We used the SLEAP key point detection framework (Pereira, Tabris et al. 2022), another software package popular with animal behavior researchers that comes with implementations of several different network architectures (we used the default option, the UNet architecture, with default hyperparameters and 50 training epochs). Accuracy results are shown in **Response Fig. 1**. We found that the UNet architecture had comparable accuracy. Our view is that while additional architectural experiments could be expected to yield some differences in performance, the DeeperCut architecture is a reasonable baseline for demonstrating our new approaches in the paper.

Response Fig 1. Accuracy of a UNet architecture (orange) versus DeeperCut / ResNet architecture (blue), trained on 300k images and tested on 4.4k held-out images.

Comment 2. *More details regarding the neural network could be helpful: What is the runtime for single frame inference? Is it possible to run the pose estimation in real-time (esp. regarding the feedback experiment for tuning the parameters)? What is the inference resolution of the neural network, i.e. the spatial localization accuracy of the landmarks, which is usually lower than the processed camera image? What are the training details of the neural network (deep learning framework, learning rate, batch size, online augmentation, etc.), or is everything kept the same as in the DeeperCut [30] code?*

The details of the neural network, including information like runtime, inference resolution, and the training procedure, are the same as in the DeeperCut paper (Insafutdinov, Pishchulin et al. 2016). The stride of the neural network was 8, yielding output heatmaps and refinement maps 8-10 times smaller in each dimension than the input images (for example, the network produces 86 x 86 square output maps for the 848 x 848 square training images). For the real-time feedback experiments, the average throughput of the neural network was approximately 10 frames per second. Other work has shown that with various optimizations, much greater throughput (100 frames per second) is also possible with the DeeperCut / DeepLabCut architecture (Kane, Lopes et al. 2020, Sehara, Zimmer-Harwood et al. 2021), although we did not find this necessary. We have clarified these points in the manuscript on **pages 37 and 39**.

Comment 3. *Sec. 2: Are there size limitation for applying the dye? Is it possible to label each digit of the rodent hand?*

The only real size limitation is the skill of the experimenter in applying the dye and the resolution of whichever cameras are used for capturing images. In preliminary experiments not shown in the paper we were able to reliably label a single digit of the rodent without dye bleeding over to other digits. We now highlight these points on **pages 29-30**.

Comment 4. *Sec. 2: The proposed camera dome allows capturing different viewpoints simultaneously. Does this generalize to arbitrary camera positions, or do the positions still need to be close to the camera poses of the dome?*

In principle, the cameras can be placed anywhere in the dome and one can increase the number of cameras beyond the eight we used in our study. The camera angles we used were chosen to maximize angular diversity while minimizing occlusions of the hand.

Moreover, the challenge test set we describe represents a diversity of camera angles not seen during training, so the evaluation results on that dataset can be taken as representative of average performance over many of the camera angles most commonly seen in practice. We have now quantified the number of different camera angles and the number of images from each angle in the challenge set in **Response Table 1** below.

Comment 5. *Sec. 2.2: How does the network generalize to different mice, i.e. gender, age, race, fur color, etc.? This is essentially related to a leave-one-subject-out validation.*

The challenge test set we describe represents male and female mice (C57BL/6) performing five different behaviors in ten different visual environments (including both monochrome and color) across two laboratories, with large variety in animal scale and resolution. We include a new **Response Table 1** below that summarizes some of the diversity represented in the challenge set, providing a sense of the

generalization performance across a sample of different mice. We now clarify these points on **pages 16 and 36**.

Animal scale (approximate):	25 px/cm	50 px/cm	100 px/cm							
Number of images:	5600	5200	11562							
Animal behavior:	String pull	Treadmill	Beam	Water reach	Pellet reach					
Number of images:	3600	3600	2400	3200	9562					
Light & camera setup:	#1	#2	#3	#4	#5	#6	#7	#8	#9	#10
Number of images:	2400	1200	3600	2400	800	800	1600	1972	6390	1200
Animal behavior:	SP	SP	T	B	WR	WR	WR	PR	PR	PR
Number of animals:	2	1	1	1	1	1	2	5	3	1
Camera type:	monochrome	color (RGB)								
Number of images:	17562	4800								

Response Table 1. Challenge set image diversity quantification. Each pair of rows quantifies the diversity of the challenge set with respect to a particular dimension. Animal scale refers to the apparent size of the animal in each image. Animal behavior refers to the different behavioral tasks the animal performed. Light and camera setup refers to the configuration of lights and the camera position from which the video was captured. Camera type refers to the pixel format of the camera used to capture the video.

Comment 6. *Sec. 2.3: The authors mention to "label one hand on the mouse" in the serial labeling use-case. It could be more practical to derive labels for all limbs or at least both hands. Is this possible with that approach?*

It is possible to label multiple landmarks one after another with the serial labeling pipeline. We now describe **new experiments** in which we trained a single network to detect hand and foot landmarks using serial labeling and an interleaved training approach described in detail in response to **Reviewer 1, Comment 2**, in a **new Supp. Fig. 4**, and in the text of the manuscript. In principle, it would also be possible to label multiple landmarks simultaneously with the serial labeling pipeline if the different landmarks were labeled with dyes that fluoresce at different wavelengths. However, to date we have only found one dye compatible with triphasic illumination (the dye must have a temporal decay constant of approximately 1-5 ms). We discuss this point on **page 32**.

Comment 7. *L107: The authors denote that they are able to label "nearly any landmark", which sounds verbose. What are the actual limitations?*

We have rephrased the original sentence on **page 6**. We also agree that a more accurate description of the limitations would be useful, which we now provide on **pages 29-30**. "Some caveats to consider are that the dye must be compatible with the subject and the illumination intensity for dye excitation must be scaled up in proportion to the square of the distance to the subject. The size of the dye region must also be large enough to be resolved consistently by the cameras. The smallest region we have labeled is a single digit on the hand of the mouse, but in principle it may be possible to resolve smaller regions."

Comment 8. *Fig. 4: The figure shows the trajectory of the landmark and the ground truth in order to visualize the accuracy. How does this capture the 2D landmark location in the image with this 1D plot? It would be more appropriate to plot the Euclidean error between prediction and ground truth as curve, in order to show the accuracy and not both curves superimposed.*

We thank the Reviewer for the helpful suggestion – we have now changed **Figure 4b** to show Euclidean error over time.

Comment 9. *Sec. 2.6: Why was SIFT chosen, as it is known to lack repeatability on almost texture-less surfaces such as the palm/digits? How does the visual bar-code react to non-rigid deformation? The used homography assumes local planarity. Would it make more sense to use an optical flow-based method, e.g. [Liu et al.: SIFT Flow: Dense Correspondence across Scenes and Its Applications, Transactions On Pattern Analysis and Machine Intelligence, 2011], for tracking the points compared to the proposed homography?*

The Reviewer raises some important points. Dense matching in textureless regions is hard for all matching methods, SIFT included. Our approach is to add fluorescent texture to these otherwise textureless regions, allowing feature-trackers to work better (including but not limited to SIFT). For our hand dataset, the local planarity assumption produced good results, and as the Reviewer states, optical flow would be another reasonable way to propagate label information between frames. The downside of optical flow is that it is not typically robust to large occlusions and out-of-plane rotations, which are both common in our data. For example, the local-homography approach can handle situations where an entire digit is hidden, whereas optical flow techniques are typically designed to handle only small occlusions near object boundaries.

Comment 10. *Sec. 2.7: How are the 50 different landmarks represented in the machine learning method (DeeperCut)? What is the benefit over using the centroid only? How does the over-parameterized pose with 50 landmarks compare to per-pixel classification, i.e. semantic segmentation?*

The 50 landmarks are represented as independently tracked points – the neural network produces a localization heatmap and refinement map for each one. One can imagine potential benefits to tracking many individual landmarks rather than the centroid only or using per-pixel classification. For example, identifying specific points on the surface of the skin could be useful for mapping cutaneous receptive fields under the right experimental setup. Per-pixel semantic segmentation could be used to classify each pixel as belonging to a particular finger segment or even neighborhood, but would be less precise for identifying the location of specific points on the skin, as would be needed for receptive field mapping (e.g.).

Comment 11. *Sec. 2.7: The authors describe the manual method for parallel labeling in great length, however, the fully automatic method is more relevant to the actual use-case and it could be sufficient to describe that solely. Does it make sense to save space and significantly shorten the manual method for parallel labeling?*

We appreciate the Reviewer’s suggestion, but we decided to retain the description of the manual method because, in practice, it is common for experimenters to have a particular set of landmarks that they want to track. For example, in research investigating manual dexterity, it is common to track the segments of the

hand/fingers. We felt it was worth the brief additional description to highlight use-cases when desired landmarks are known and can be manually defined.

Comment 12. *Sec. 2.7. From the text, it is unclear how the regions/centroid/clusters are initialized for the automatic method. Are the initial locations random and then further clustered?*

Yes, a set of seeds are selected randomly but with a minimum-distance requirement to ensure they are spaced apart. The points within a certain radius of each seed are treated as a cluster. Therefore, some points belong to multiple clusters. We have now clarified this point on **pages 41-42**.

Comment 13. *L477. The authors propose a semi-automatic method for the dense labeling problem, "because a human annotator cannot be given a simple description of each landmark". However, such a manual labeling was done e.g. in [Guler et al., Densepose. Dense human pose estimation in the wild, Conference on Computer Vision and Pattern Recognition, 2018].*

We thank the Reviewer for pointing out our oversight – we have changed the language in the manuscript to reflect the fact that this process is challenging but not impossible (**page 27**).

Comment 14. *L654. The authors marked the landmark as present only if "the annotators' labels were within 5 pixels of each other". This could create false negatives, and as the authors denote before that images without labels help boost the model performance. Did the authors notice any drawbacks by using this strategy?*

The main drawback of requiring inter-annotator agreement is that the number of images that successfully receive labels is fewer, and thus more labeling is required. The Reviewer also correctly points out that model accuracy can improve somewhat – however, as the labels removed tended to involve cases where the true landmark position was ambiguous, there was not an obvious way to fix the problem besides excluding them from consideration.

Comment 15. *Sec 4.4. The authors mention an "image pre-scaling factor of 0.8". How does that change the image, or is this a simple resize operation? Further, the authors mention a "conversion of color images to gray-scale with 50% probability". Why is that done, and why do the authors not provide a separate model for color and gray-scale? How is the 1-channel gray-scale (intensity) image converted to a 3-channel color (RGB) image? Why is so much attention spent on red illumination (e.g. Supp. fig. 1)?*

The image pre-scaling factor refers to a simple resize operation. This value was selected by a grid search over a few different parameter values. The pre-scaling factor serves to select a receptive-field size for the neural network. The need to handle both color and grayscale was dictated by the presence of both types of image data in the challenge set and by the fact that both are commonly used by labs, so we wanted to maximize generalizability. Grayscale images were converted to 3-channel color images by simple channel-wise duplication, after which they were processed by the augmentation pipeline. We used red illumination because it is common in rodent experiments to shift animals' light/dark cycle because mice perform behaviors more reliably when it is "night". Mice cannot see in the red portion of the visible spectrum, and thus, the circadian shift is not affected by red light. We now clarify this point on **page 33**.

Comment 16. *References: Please use a consistent format. Esp. [13,38,41] consider replacing bioRxiv with proper conference/journal publication, [14,15,40] pages are missing, [19] pages contain strange string, [26,28,31] for conference use consistent usage of pages or not print pages at all, [28,31,39,43,61] avoid an additional conference abbreviation, [30,60] citations are incomplete, etc.*

We thank the Reviewer for identifying these formatting errors generated by the reference management software. We have corrected inconsistencies and replaced the bioRxiv references with their subsequent journal publications.

Comment 17. *Supp. fig. 6: The authors mention the use of "morphological erosion". However, the images look non-binary, but the morphological operations for foreground/background segmentation are only well-defined on binary images, esp. when blending in different backgrounds. Please clarify.*

The underlying images are binary, but the rendering method we used made the images in the original figure appear to be non-binary. This has been corrected in what is now **Supp. Fig. 7**.

Minor points

We thank the Reviewer for the comments below and have made all requested changes and corrections (Comments 18-26).

Comment 18. *L164: Consider using consistent naming: "signal to noise" -> "signal-to-noise ratio" as in L149*

Comment 19. *L171: Instead of "islands" consider using "blobs" as the more relevant image processing term.*

Comment 20. *L181ff: Capitalization "section" - "Section".*

Comment 21. *L208: Typo "unform" - "uniform".*

Comment 22. *L456: The punctuation is confusing when listing the errors of the different training. Please check usage of "," and ";".*

Comment 23. *L510/513: Consider replacing "many-camera setup" with "multi-camera setup" to be more consistent with the remaining text.*

Comment 24. *L659: There are Sections 4.4/4.6, but no 4.5!*

Comment 25. *L936: "Video (10x speed)" suggests a fast-forward video, but actually it is slowed down.*

Comment 26. *Supp. video 1: The labels of the camera feed "UV" and "visible" seem swapped.*

Comment 27. *Supp. video 3: What do the numbers next to the detected landmark denote?*

The numbers next to the detected landmark denote confidence score. We have added an explanatory title screen to the video and a more detailed explanation to the legend.

Comment 28. *Supp. video 5: Without reading the paper beforehand, the dots are not self-explanatory and a short explanation could help the reader.*

We thank the Reviewer for the comment and have added an explanatory title screen to the video and a more detailed explanation to the legend.

Reviewer 3

The authors describe two approaches to automatically label keypoints on an animal in video frames to serve as ground truth for pose estimation algorithms. In the first approach, they paint a single landmark (a mouse hand) using UV fluorescent probe, and use an 8 camera setup, multiple UV light sources, and a triphasic data collection regime to record the position of the painted region with high signal to noise. They demonstrate that this approach can be used to collect a reasonably large training dataset (~300 k frames), across three different behaviors, and that the hand can be tracked using a model trained on the diverse training dataset, but not one from a single camera view. They then show that the diverse training dataset produces a moderate-quality hand tracker on out-of-domain images, after applying some additional algorithmic refinement by resizing the input images by scale factors optimizing model confidence. In the second approach, they apply the fluorescent label to the human hand, essentially as an added source of texture, and present several strategies for training keypoint detectors to locate regions of interest in these blobs.

Producing more generalizable pose tracking models is important for the life sciences, and automated ways of collecting training data are a key step in that direction. I think the approach of UV labels is clever, and I like the half-dome optical setup and approach for illumination randomization. I think the result showing the enhancement of tracking with added labels is useful, and there are multiple technical details, e.g. clip-scale optimization that others in the field may find useful. While I'm not sure that fluorescent labeling will become widespread in the field, I do think this manuscript contains multiple innovations that people in the field will find useful. That said, I do think the manuscript suffers from several weaknesses. First, I think several of the claims are overstated, mostly in regards to generality of tracking from models. Second, while the manuscript claims the serial tracking approach can be used to track multiple points, evidence is not provided. Third, the parallel tracking approach is comparatively under-contextualized and benchmarked, and I am not convinced of the results. Overall, I am not against publication of this manuscript, but it needs revision.

Comment 1. *What is claimed: L497: "Together, the generalizability and scalability of these approaches enable automated pose detection and kinematic quantification across species with less effort, better accuracy, and higher resolution than is feasible with manual annotation approaches." "Together, these results demonstrate that [...], and that our fluorescence labeling approach can provide the necessary quality and quantity of training data." I don't believe these are supported by the data. I do think that the advances in the paper are useful and meaningful to the field. But the performance of the generalized rodent*

hand tracking in figure 4e falls short of the manual annotation approach in 3g (I realize these are different test domains). I do think that these approaches can facilitate generalization and perhaps less need for fine tuning, but I don't think it replaces them. Same comment for accuracy and resolution. Similarly, the ability of this approach to perform pose tracking has not been established (only a single point shown). There are multiple comments about kinematic quantification in the manuscript, but the only example is figure 4b, which is more a demonstration of error reduction.

We thank the Reviewer for this feedback. We agree that our method does not replace manual annotation for all tasks. Indeed, as we show in **Fig 3g** (and as many others have shown), a modest amount of manual annotation is more than sufficient for high-quality tracking *if* the visual environment does not differ from that of the training data. However, we believe we have shown that due to improved generalization and automated collection of massive amounts of training data that are free of the frame-by-frame variability of manual labels, our method has achieved a new point on the what could be called the accuracy-generalization Pareto front. To more accurately describe this point, we have edited the text on **page 28** to now read “Together, the scalability of these approaches enables automated markerless tracking with less effort and greater generalizability to new visual environments than is typically feasible with manual annotation approaches.”

Regarding the results in **Fig 4e** as they compare to those in **Fig 3g**, we would like to highlight a few points.

- 1) The challenge test set used in **Fig. 4** is just that – intentionally quite challenging and very different than the extremely uniform test images used in **Fig. 3g**. The challenge test represents male and female mice (C57BL/6) performing five different behaviors in ten different visual environments (including both monochrome and color) across two laboratories, with large variety in animal scale and resolution. Moreover, the water reach, treadmill, string pull, and balance beam behaviors were captured on behavioral apparatuses different from those represented in the training data. We include a new **Response Table 1** above that summarizes some of the diversity represented in the challenge set, providing a sense of the generalization performance across a sample of different mice.
- 2) The difficulty of these challenge images for trained networks is reflected by the very poor performance of a network trained with only 1000 diverse images (green) in **Fig. 4e**. Moreover, the performance of the small-scale regime networks from **Fig. 3g** (trained from a uniform set of images, as is typical with manual labeling) is far worse still. We originally only showed the precision-recall curves and pixel errors for these small-scale models in **Fig. 3h**, where their performance substantially degrades (to an unusable level) even when tested in an environment that is easier than the challenge test set (diverse test images acquired from the dome). We now add demonstration of the further reduced performance of these models when tested on the challenge set to a **new Fig. 4e**, also shown below in **Response Fig. 2a**. Stated briefly, comparing the good performance of these small-scale regime networks in **Fig. 3g** to their very poor performance in **Fig. 4e** is precisely why new methods for generating large, diverse training sets are needed.
- 3) If we remove images from the challenge set that had particularly poor resolution or image scales that were dramatically different than the training data, performance improves even more (**Response Fig. 2b**). Thus, to achieve top performance a typical user would just need to ensure that their experimental data at least moderately resembles the scale and resolution of their training data (even if lighting conditions, camera angles, etc. all differ substantially). That said, for presentation in the paper we did not feel that removing images from the challenge set best reflected our point – that even with

extremely challenging images, our approach far outperforms what could feasibly be achieved with manual labeling.

4) We also provide additional ways for a user to improve performance, including further data augmentation, publicly available temporal smoothing approaches, and our real-time optimization during data collection (Supp. Fig 5b).

Thus, we feel that what we present does accurately support our (now edited) statements.

With respect to quantifying kinematics, we opted not to include these types of results in the manuscript as they are essentially a natural and easily achievable result once high-quality tracking is achieved. Prior work such as Anipose, DLC, and SLEAP have developed valuable resources for pose estimation and kinematic tracking, and the approaches we describe here can generate markerless tracking data that can easily feed into these pipelines. We have removed mention of kinematics and pose estimation in the text when it is not relevant.

Response Fig 2. Additional quantification of challenge set performance. a) Precision-recall curves (left) and pixel error quartile plots (right) for the *challenge* set, expanded to include results for the small-scale regime models in Fig. 3g (gray, light blue, dark blue). b) Same plots as (a), but restricted to a subset of the challenge set in which scale and resolution are approximately matched to training set.

Comment 2. *Serial tracking approach. The authors haven't demonstrated serial application. I think this is likely feasible, but it needs to be shown. It is possible that there will be e.g. swaps across the hands and feet.*

We thank the Reviewer for this suggestion, which is in line with other Reviewers' comments. We now describe **new experiments** in which we trained a single network to detect hand and foot landmarks using serial labeling and an interleaved training approach, described in detail in response to **Reviewer 1, Comment 2**, in a **new Supp. Fig. 4**, and in the manuscript text.

Comment 3. *Contextualization, exposition, and benchmarking of the parallel tracking approach. This work was comparatively underdeveloped to the serial tracking approach. There is a schematic algorithm, and some quantification, but the 1) contextualization within computer vision 2) exposition and 3) quantification is lacking.*

Comment 3a. *What is the relationship between this approach and other approaches in computer vision for automated keypoint discovery. You can imagine taking the same approach as used here, but using textural features or high contrast features directly from CV. Work in stereopsis has pursued this (e.g. KeypointNet) and likely monocular CV. Moreover, in the domain of hand tracking there are approaches that use motion*

capture as automated labels (e.g. Shangchen Han,...,Robert Wang, ACM transactions Graphics 2020). Moreover, some of the work with automated neighborhood selection touches on surface tracking and it would be good to at least acknowledge this connection.

We thank the Reviewer for these suggestions and have now added additional citations and explained more of the prior work in depth on **page 31**.

Comment 3b. *Can you include a workflow of the algorithm.*

We have now added a workflow of the parallel tracking approach as a **new Supp. Fig. 9**.

Comment 3c. *How does this SIFT approach compare with using features from computer vision within the same pipeline. Is it a strength that this does not e.g. generalize to tracking on the top of the hand?*

We would like to offer some clarification. SIFT features were detected on the top of the hand and could have been used for matching. The reason the back of the hand appears devoid of matches in **Fig. 5** is that for visualization and demonstration purposes we only used templates that covered the palm surface of the hand, and thus only features on the palm side are colorized. The same procedures could be applied to the top of the hand or the entire hand depending on which template images are chosen. We now clarify this point on **page 23**. Other computer vision features were not evaluated because of the good performance of SIFT matching, but it is possible that other features, for example those learned from data, could provide additional robustness.

Comment 3d. *There needs to be some sort of yardstick baseline/sanity check for these approaches that can give a sense of the significance of the precision-recall curves in 6e,6k. These curves give me very little sense of the true performance, as they depend heavily on the types of images in the test set, which could be very in-domain and a simple e.g. regression approach based on CV features might do just as well. This approach certainly doesn't have broad generalization across hand pose, image scale, etc. (the reason for many... [author's note: the Reviewer's comment appears to have been cut off])*

We thank the Reviewer for the reasonable critique. Taken as a whole, the parallel tracking section of the paper is not meant to claim that the hand tracker presented is superior to state-of-the-art models with respect to out-of-domain generalization. Rather, our intent was to demonstrate a new method for producing dense ground-truth data. To that end, we showed: 1) that fluorescent dye applied in a random pattern can be used to produce trackable features, 2) that trackable features, even if not matched in every image, can be pooled together to produce reliable labels for tracking, and 3) that an existing landmark-tracking neural network architecture can be trained to track many of these fluorescence-derived landmarks simultaneously.

The reason for including the performance curves in **Figs. 6e** and **6k** was to compare different internal configurations of the label-generation pipeline, showing the effect of number of templates and homography reprojection, respectively. In essence, tracking accuracy is being used as a proxy to measure the relative strength of different algorithms for producing labels from the raw fluorescence data, which we demonstrate in **Fig. 6**.

The question of CV features is an interesting one: are computer vision features, derived directly from an image, already sufficient to enable this type of landmark detection without applying a random fluorescent pattern? If so, the implication might be that applying the fluorescent pattern is superfluous. Certainly, with respect to SIFT features, the random pattern is important – we found that without the random speckle pattern, the average number of SIFT features detected per image is negligible (**Response Fig. 3**). This is presumably due to the lack of visual texture on the human hand.

Response Fig. 3. The average number of SIFT features per image drops dramatically without the addition of the random fluorescent speckle pattern.

There is also a larger point to make – even if a totally different machine learning pipeline (such as CV features plus regression) could produce strong trackers in this domain, one would still need significant quantities of ground-truth data for training and evaluating such a model on so many landmarks. We show that random fluorescent labeling provides a novel way to obtain that ground-truth data easily and in parallel, at scale.

Comment 3e. *L485 ‘Using this approach, our model could reliably identify dozens of landmarks on a hand’. don’t believe the claim of 50 simultaneously tracked keypoints is every quantified. Given the keypoint error is ~40 px (Figure 6k) I am not sure there is enough space on the hand/finger to simultaneously detect such a large number of keypoints. Given the large amount of jitter in the supplemental video, I don’t believe it is accurate to say these are uniquely identified.*

The lack of scale bars mentioned by the Reviewer in the next comment, may have contributed to the impression that 40 px accuracy is not sufficient to distinguish individual points. With the addition of scale bars to **Fig. 6**, we can see that each finger is ~150 px wide. Thus, 40 px is sufficient resolution to resolve 34 points across the width of the finger, which is approximately the number seen in **Fig. 6h,j,l**.

However, the Reviewer brings up another point, which is the issue of jitter. We think this jitter has more to do with the neural network’s resolution limits than the underlying fluorescence data. The heatmap produced by the DeeperCut network has finite resolution – based on experiments we did not show in the paper, the jitter appears to be caused by the aliasing introduced by the heatmap rather than by the underlying training data. Specifically, changing the heatmap scale changes the scale of the jitter by a corresponding amount. Thus, we believe a fair interpretation is that higher resolution architectures, perhaps with greater GPU memory, could reduce this type of jitter. Since the focus of the paper is on the fluorescent labeling approach, we feel that architectural considerations are somewhat orthogonal to our main claims.

Comment 3f. *Nits: scale bars on the images are needed (how big are 40 pixels?). I couldn’t find a description of the deep learning model used for hand tracking here, hyperparameters, etc.*

We have now added scale bars in units of pixels and millimeters to **Fig. 6d,f,j,l**. Regarding the deep learning model used for hand tracking, the same model architecture was used for the human hand as was used for the mouse (DeeperCut, default hyperparameters, plus our data augmentation approach described in Methods). We have now made this explicit on **page 37** of the manuscript.

Comment 4. *Generalization in the challenge dataset: How well does the approach generalize to other behaviors (eg rotarod?). Many of these domains are quite similar behaviorally.*

The challenge test represents male and female mice (C57BL/6) performing five different behaviors in ten different visual environments (including both monochrome and color) across two laboratories, with large variety in animal scale and resolution. Moreover, the water reach, treadmill, string pull, and balance beam behaviors were captured on behavioral apparatuses different from those represented in the training data. We include a new **Response Table 1** above that summarizes some of the diversity represented in the challenge set, providing a sense of the generalization performance across a sample of different mice.

While rotarod was not explicitly tested, we feel our results strongly suggest the approach will generalize to other behaviors as long as landmarks of interest are clearly visible.

Minor points

Comment 5. *Illumination randomisation vs. other image augmentation during training*

We varied actual illumination (using the multi-light dome), but also simulated other illumination changes *in silico* in our augmentation pipeline (contrast, color, and intensity – please see **page 37**).

Comment 6. *L29 “generating hidden labels free of human error using fluorescent markers”. Humans choose the position of labels, so these are very dependent on human choices.*

We have revised this sentence to emphasize only the elimination of frame-by-frame variability of human labels.

Comment 7. *What body structures can this approach be applied to? Is it only skin? There are some comments about fur in the conclusion but I do not believe the limitations of the existing approach were clearly stated.*

We have successfully applied the dye to both skin and fur. We now clarify this point on **page 6**. We also agree that a more accurate description of the limitations would be useful, which we now provide on **pages 29-30**. “Some caveats to consider are that the dye must be compatible with the subject and the illumination intensity for dye excitation must be scaled up in proportion to the square of the distance to the subject. The size of the dye region must also be large enough to be resolved consistently by the cameras. The smallest region we have labeled is a single digit on the hand of the mouse, but in principle it may be possible to resolve smaller regions.”

Comment 8. *The manuscript but feels very long. Some of the technical details could be moved to supplement. The parallel tracking section especially.*

We appreciate the Reviewer’s suggestion. We have moved some information to the Methods. For the remaining text, we feel for many readers, especially those who might want to apply these methods but may not have extensive background knowledge, the main text provides important explanation.

Comment 9. *The comparison of manual vs fluorescence labels in supp figure 3a is surprising. I would imagine that there are a subset of frames that are visible in RGB, but lack SNR in dye to get detected (e.g. due to self-occlusion of the UV source). What fraction of frames is this and is the test set here randomly sampled or restricted to frames in which there is UV ground truth?*

The precision-recall curves in **Supp Fig. 3a** (left panel) were performed on a randomly sampled subset that included frames with and without UV ground truth, while the pixel error quartile plots (right panel) were performed only on frames with UV ground truth (since pixel error cannot be computed without a ground truth label). Of the 600 evaluation frames, 58.5% had both a manual label and a UV label, while 19.3% had neither a manual label nor a UV label (likely indicating that the landmark was hidden from view), for a total of 77.8% agreement. The remaining 22.2% of evaluation frames had one or the other: 12.5% had a UV label but no manual label, while 9.6% had a manual label but no UV label.

One interpretation for why the two ground truth datasets had significant areas of non-agreement while still producing similar evaluation results is that neither ground truth dataset is strictly “harder” than the other – due to their discrepancies they have different “hard subsets” on which the models tend to do poorly, but the size of these subsets are roughly equal. In other words, while UV images will occasionally fail to produce a label, the same is true of manual labeling, and the net result is similar performance metrics when using either set of ground truth.

To the Reviewer’s question regarding images in which the UV dye fails to produce a label due to low SNR, these tend to be images in which the landmark has poor visibility and for which the models will produce low confidence scores. Thus, the models will likely not be penalized too much (in terms of AUC) for producing a “false positive” (really an erroneously missing ground truth label) on these images.

Comment 10. *Section 2.3, l237. The term ‘manual’ here is confusing, since these are not actually manually labeled.*

We agree that the term is not ideal. We originally used the two-word phrase “manual regime” whenever we were referring to a small number of labels, but ones that were in fact automatically generated. We have now changed this to “small-scale regime”.

Comment 11. *Figure s3b – were the number of frames here balanced across conditions (e.g. of lighting/viewpoint).*

In the top panel of **Supp. Fig. 3b**, the subsets of images were selected randomly and therefore the number of frames was not balanced across conditions. In the middle and lower panels, however, the number of frames was inherently balanced to within 0.4% because the number of frames for each lighting/viewpoint combination was nearly the same.

Comment 12. *Diverse dataset pretraining. How does pretraining on other action datasets, e.g. COCO compare to the rodent dataset?*

We have not pre-trained on other action datasets, but this is a good suggestion for future work.

Comment 13. *Figure 4: Will clip level optimization increase the false positive rate of the approach? Do all images in the test set contain hands?*

Not all images in the test set contain hands. Therefore clip-level optimization could potentially increase the false positive rate of the approach for a specific confidence threshold. However, all points of the precision-recall curve for clip-level optimization are above those of the curve without it, so there always exists a confidence threshold for which the false positive rate is lower (at least for the datasets we have produced).

Comment 14. *Why stop at 300 k frames in the diverse training set?*

We decided that we had reached a sufficient number of training examples for the needs of establishing and refining these methods. That said, given the ease with which data can be collected, much larger datasets could be collected, though at some point, image diversity, not quantity, is likely to be the limiting factor for performance.

Comment 15. *L399 ‘optimal’ number of templates – is there justification for why 10, and not 20, 30, 40, etc. is optimal?*

The language is again not ideal. The intended meaning was “an optimal *set* of templates given that a certain number will be selected”. This is similar in spirit to the k -means algorithm, where k is specified as input to the algorithm. However, the greedy algorithm is not necessarily optimal even in this weaker sense. We have clarified this point and changed the language in the manuscript on **page 23**: “To help select a set of template images that achieve good matching coverage and are not too redundant (i.e., templates containing a diversity of poses with the 12 digit segments mostly visible), we devised a greedy algorithm that...”

Comment 16. *L436 – a diagram of the network architecture would help clarify the training procedure*

The original DeepLabCut paper provides this diagram (Mathis, Mamidanna et al. 2018), while the DeeperCut paper provides details on the training procedure (Insafutdinov, Pishchulin et al. 2016). We hope that by referencing these papers throughout the text, the curious reader can find details on the network architecture.

With these extensive changes, and the inclusion of considerable additional new data and analyses, we feel we have answered the Reviewers’ helpful comments in detail.

With best regards,

Eiman Azim

Associate Professor, William Scandling Developmental Chair, Molecular Neurobiology Laboratory, Salk
Institute for Biological Studies

Associate Adjunct Professor, UCSD Division of Biological Sciences, Section of Neurobiology

eazim@salk.edu

(858) 453-4100 x1074

<http://azim.salk.edu/>

<http://www.salk.edu/scientist/eiman-azim>

References

- Insafutdinov, E., L. Pishchulin, B. Andres, M. Andriluka and B. Schiele (2016). "DeeperCut: A Deeper, Stronger, and Faster Multi-person Pose Estimation Model." European Conference on Computer Vision.
- Kane, G. A., G. Lopes, J. L. Saunders, A. Mathis and M. W. Mathis (2020). "Real-time, low-latency closed-loop feedback using markerless posture tracking." eLife **9**: e61909.
- Kim, I., Y. Kim and S. Kim (2020). "Learning loss for test-time augmentation." Advances in Neural Information Processing Systems.
- Krizhevsky, A., I. Sutskever and G. E. Hinton (2017). "Imagenet classification with deep convolutional neural networks." Communications of the ACM **60**(6): 84-90.
- Mathis, A., P. Mamidanna, K. M. Cury, T. Abe, V. N. Murthy, M. Mathis and M. Bethge (2018). "DeepLabCut: markerless pose estimation of user-defined body parts with deep learning." Nature Neuroscience **21**: 1281-1289.
- Pereira, T. D., N. Tabris, A. Matsliah, D. M. Turner, J. Li, S. Ravindranath, E. S. Papadoyannis, E. Normand, D. S. Deutsch and Z. Y. Wang (2022). "SLEAP: A deep learning system for multi-animal pose tracking." Nature Methods **19**(4): 486-495.
- Sehara, K., P. Zimmer-Harwood, M. E. Larkum and R. N. S. Sachdev (2021). "Real-Time Closed-Loop Feedback in Behavioral Time Scales Using DeepLabCut." eNeuro **8**(2).
- Shanmugam, D., D. Blalock, G. Balakrishnan and J. Guttag (2021). "Better aggregation in test-time augmentation." IEEE International Conference on Computer Vision.

REVIEWERS' COMMENTS

Reviewer #1 (Remarks to the Author):

The authors have gone above and beyond to address the reviewers concerns as well as strengthened the manuscript considerably. Clarifications, new data as well as extended figures have helped the clarity of the manuscript.

The reviewer has no outstanding comments and congratulates the authors for their wonderful manuscript.

Reviewer #2 (Remarks to the Author):

First and foremost, I would like to thank the authors for replying to the comments and remarks of the reviewers. As disclosure, I already reviewed the initial submission of the submitted manuscript. The authors improved the quality and content of the paper and addressed all of my initial concerns. Overall, the paper is now in good shape for publication, and provides a valuable addition to the field of animal behavior analysis.

Minor comments left:

- Supp. Fig. 4: typo "hindlimbs"
- L686: In Supp. Fig. 4 the network training loss is denoted as "loss function" but in the text it states "objective". For clarity, consider using the same wording.

Reviewer #3 (Remarks to the Author):

I thank the author's for their close attention to my and all the reviewers' comments. The manuscript and its revision represent a tremendous amount of work and I am again thankful for the efforts on the part of all involved in putting it together. I think the manuscript contains conceptual and technical advances that will be useful to many in the field, and I am mostly happy with it.

I however, still have an issue with the claims regarding parallel tracking, which I believe are overstated and I do not think the manuscript can be published with these claims as written. I want to be clear that I think the parallel tracking work is interesting and has some merit. But the benchmarking is cursory relative to claim of dozens of simultaneously tracked points. I think this can be handled through text changes only -- I don't want these to lead to significant delays in publishing the manuscript.

A claim in the abstract is the following:

"a technique for massively parallel tracking of hundreds of landmarks simultaneously using computer vision feature matching algorithms, providing dense coverage for movement analysis at a resolution not currently available"

In the supplemental videos there are also claims of tracking "50 unique landmarks"

In the text

L463 "Using this approach, our model could reliably identify dozens of landmarks on an unlabeled hand"

I have reservations that this 'tracks' 50 unique landmarks and that this level of resolution is not currently available. First, this is never demonstrated or validated, and the statistics provided are underwhelming. I would not have confidence that tracking with 40 mm pixel error for points spaced 30 pixels apart yields reliable tracking. From these numbers I would guess that there are multiple times when the pattern is significantly distorted (this is also seen in the supplemental video).

Human labels are of course difficult here, but there are also more heuristic statistics that are useful. How many frames are each template detected on (from the supp. video, it seems it is very sensitive to eg orientation)? What is the template to template centroid distance? How consistent is the spatial relationship between centroids? In optics, there are well defined criteria for when two point sources are considered separated. Similarly here, you can adopt these heuristics to get a sense of how robust the 'tracking' of these points are.

wrt 'at a resolution not currently available' it should be noted that 3DMD systems achieve highly accurate 3d mesh tracking, with ~1 mm resolution (using a somewhat similar approach).

Nit: It should be made clear in the text that clip-level optimization has tradeoffs with the false positive rate.

Nit: S4c - not sure what the bar plot (boxplot?) definitions are wrt median/mean/ci.

We are enclosing a revised version of our manuscript “Large-scale capture of hidden fluorescent labels for training generalizable markerless motion capture models”.

We include below a detailed account of the specific responses to each of the Reviewers’ comments in the order they were made, and all changes in the original manuscript are tracked.

Reviewer 1

The authors have gone above and beyond to address the reviewers concerns as well as strengthened the manuscript considerably. Clarifications, new data as well as extended figures have helped the clarity of the manuscript.

The reviewer has no outstanding comments and congratulates the authors for their wonderful manuscript.

Reviewer 2

First and foremost, I would like to thank the authors for replying to the comments and remarks of the reviewers. As disclosure, I already reviewed the initial submission of the submitted manuscript. The authors improved the quality and content of the paper and addressed all of my initial concerns. Overall, the paper is now in good shape for publication, and provides a valuable addition to the field of animal behavior analysis.

Minor points

Comment 1. *Supp. Fig. 4. typo "hindlimbs".*

The typo has been corrected.

Comment 2. *L686. In Supp. Fig. 4 the network training loss is denoted as "loss function" but in the text it states "objective". For clarity, consider using the same wording.*

We thank the Reviewer for the comment and have changed “loss function” to “objective function”.

Reviewer 3

I thank the authors for their close attention to my and all the reviewers' comments. The manuscript and its revision represent a tremendous amount of work and I am again thankful for the efforts on the part of all involved in putting it together. I think the manuscript contains conceptual and technical advances that will be useful to many in the field, and I am mostly happy with it.

I however, still have an issue with the claims regarding parallel tracking, which I believe are overstated and I do not think the manuscript can be published with these claims as written. I want to be clear that I think the parallel tracking work is interesting and has some merit. But the benchmarking is cursory relative to claim of dozens of simultaneously tracked points. I think this can be handled through text changes only - I don't want these to lead to significant delays in publishing the manuscript.

A claim in the abstract is the following: "a technique for massively parallel tracking of hundreds of landmarks simultaneously using computer vision feature matching algorithms, providing dense coverage for movement analysis at a resolution not currently available"

In the supplemental videos there are also claims of tracking "50 unique landmarks"

In the text (L463) "Using this approach, our model could reliably identify dozens of landmarks on an unlabeled hand"

I have reservations that this 'tracks' 50 unique landmarks and that this level of resolution is not currently available. First, this is never demonstrated or validated, and the statistics provided are underwhelming. I would not have confidence that tracking with 40 mm pixel error for points spaced 30 pixels apart yields reliable tracking. From these numbers I would guess that there are multiple times when the pattern is significantly distorted (this is also seen in the supplemental video).

Human labels are of course difficult here, but there are also more heuristic statistics that are useful. How many frames are each template detected on (from the supp. video, it seems it is very sensitive to eg orientation)? What is the template to template centroid distance? How consistent is the spatial relationship between centroids? In optics, there are well defined criteria for when two point sources are considered separated. Similarly here, you can adopt these heuristics to get a sense of how robust the 'tracking' of these points are.

W.r.t. 'at a resolution not currently available' it should be noted that 3DMD systems achieve highly accurate 3D mesh tracking, with ~1 mm resolution (using a somewhat similar approach).

We thank the Reviewer for the comments and for highlighting places in the text where clarification would be helpful. We have rephrased the claim in the abstract to instead refer to “many” landmarks rather than “hundreds”, as the number of landmarks tracked is up to user discretion and will be highly dependent on cameras, density of labeling of the subject, etc. Thus, while we believe “hundreds” of landmarks are easily attainable in our data (for example, if automatic neighborhood selection is used across the entire hand, rather than just the 50 landmarks on one finger, as we show), stating “hundreds” might be overly specific. We have also removed the phrase “at a resolution not currently available”. We have also clarified instances throughout the text to reflect the fact that we track *many* landmarks simultaneously, but that the precise number will depend on the neighborhood size and other application-specific factors.

With respect to the phrase “unique landmarks”, our intention was simply to indicate the number of landmarks consistently tracked across frames, not to make any claims about their statistical overlap. To be clear, the landmark identities are produced by the model, not inferred from the prediction positions – even landmarks with predictions that are close together or partially overlapping will retain their separate, “unique” identities. The definition of separation of point sources from optics, while a good suggestion, does not consider possible covariation in the positions of the tracked landmarks, complicating the picture and

potentially exaggerating the extent to which distinct points tend to overlap. Also note that we do not claim that the landmarks never partially overlap. We feel that the revised text, alongside the quantitative data presented in the figures as well as the supplemental movies, provide the reader with sufficient information to judge the accuracy of the tracking when using manual and automatic neighborhood selection on this representative dataset.

Minor points

Comment 1. *It should be made clear in the text that clip-level optimization has tradeoffs with the false positive rate.*

We thank the Reviewer for this suggestion and have added text in the discussion section, on page 26: “At the same time, heuristics like ours that optimize confidence scores may also result in higher false positive rates if not applied judiciously. For instance, the confidence threshold, if used, will need to be adjusted appropriately.”

Comment 2. *S4c - not sure what the bar plot (boxplot?) definitions are w.r.t. median/mean/CI.*

The legend has been updated to clarify the definition of the box plot (median, with 25th / 75th percentiles).

With these changes, we feel we have answered the Reviewers’ helpful remaining comments.

With best regards,

Eiman Azim

Associate Professor, William Scandling Developmental Chair, Molecular Neurobiology Laboratory, Salk Institute for Biological Studies

Associate Adjunct Professor, UCSD Division of Biological Sciences, Section of Neurobiology

eazim@salk.edu

(858) 453-4100 x1074

<http://azim.salk.edu/>

<http://www.salk.edu/scientist/eiman-azim>